# Unexpected mutual regulation underlies paralogue functional diversification and promotes epithelial tissue maturation in *Tribolium*

Daniela Gurska [1], Iris M. Vargas Jentzsch[1] & Kristen A. Panfilio [1,2 ✉]

Insect *Hox3/zen* genes represent an evolutionary hotspot for changes in function and copy number. Single orthologues are required either for early specification or late morphogenesis of the extraembryonic tissues, which protect the embryo. The tandemly duplicated *zen* paralogues of the beetle *Tribolium castaneum* present a unique opportunity to investigate both functions in a single species. We dissect the paralogues' expression dynamics (transcript and protein) and transcriptional targets (RNA-seq after RNAi) throughout embryogenesis. We identify an unexpected role of Tc-Zen2 in repression of *Tc-zen1*, generating a negative feedback loop that promotes developmental progression. Tc-Zen2 regulation is dynamic, including within co-expressed multigene loci. We also show that extraembryonic development is the major event within the transcriptional landscape of late embryogenesis and provide a global molecular characterization of the extraembryonic serosal tissue. Altogether, we propose that paralogue mutual regulation arose through multiple instances of *zen* sub-functionalization, leading to their complementary extant roles.

[1] Institute of Zoology: Developmental Biology, University of Cologne, 50674 Cologne, Germany. [2] School of Life Sciences, University of Warwick, Coventry CV4 7AL, UK. ✉email: Kristen.Panfilio@alum.swarthmore.edu

Change over macroevolutionary time scales can produce new gene functions, with the *Hox3/zen* genes of insects representing a striking example. Across the bilaterian animals, Hox genes are conserved in genomic organization, expression, and function, with roles in tissue specification along the anterior–posterior body axis of the developing embryo[1]. Instead, the *Hox3* genes in winged insects, known as *zen*, are prone to genomic microinversions[2–4], and they are required in the novel tissue domain of the extraembryonic membranes (EEMs)[5].

The EEMs, simple (monolayer) epithelia, are an evolutionary innovation to protect the developing insect. Initially they surround the early embryo, forming a multilayered barrier from the environment, with an outer serosa and inner amnion[5]. In particular, the serosa is capable of innate immune responses[6,7] and it secretes a thick chitin-based cuticle that mechanically reinforces the eggshell and provides desiccation resistance[8–11]. However, in later development the EEMs must actively withdraw to ensure correct closure of the body[12,13].

Functional studies have identified roles for *zen* in either early EEM specification or late EEM withdrawal (reviewed in ref. [14]). Although the *Hox3* locus is prone to lineage-specific duplications[15,16], to date a single EEM function—specification or morphogenesis (late tissue remodeling for withdrawal)—is known per species in bugs and flies[17–19]. This is even true in the derived case of the fruit fly *Drosophila melanogaster*, which has three functionally distinct paralogues: *zen* itself is involved in EEM specification, the duplicate *z2* is not required for embryogenesis, and the dipteran-specific *bicoid* encodes a maternal determinant with no extraembryonic role[18,20–23]. Furthermore, secondary tissue simplification of the EEMs in *Drosophila* obviated the requirement for the late withdrawal function[14]. Thus, the original role of *zen* within the extraembryonic domain has been obscured by ongoing evolutionary changes in both *zen* and the EEMs.

There is a notable exception to the pattern of a single EEM role of *zen* per species. In the red flour beetle, *Tribolium castaneum*, *zen* has undergone a tandem duplication. *Tc-zen1* was first cloned from cDNA[24], while *Tc-zen2* was later identified by sequencing the Hox cluster directly[25]. The paralogues are striking for their compact, shared gene structure and for their proximity: within the 58-kb region between *Hox2/mxp* and *Hox4/Dfd*, the paralogues occupy a <3-kb interval, with only 216 bp between the 3′ UTR of *Tc-zen1* and the initiation codon of *Tc-zen2*[25]. Nonetheless, subsequent functional diversification has equipped the paralogues with either of the two known EEM functions: early-acting *Tc-zen1* specifies the serosal tissue, while *Tc-zen2* is required for late EEM withdrawal morphogenesis[12,26]. We thus undertook a detailed molecular characterization of the beetle paralogues to elucidate the evolutionary history of *zen* functional changes, and more generally to shed light on the regulatory implications of paralogue retention and diversification after a gene duplication event.

Here we present differences in the regulation of *Tc-zen1* and *Tc-zen2* as well as in their own transcriptional signatures as homeodomain transcription factors, providing the first detailed functional dissection of insect *zen* duplicates outside of *Drosophila*. Surprisingly, peak expression does not coincide with the time of primary function—detectable morphologically and transcriptionally—for *Tc-zen2*, which despite its late role has strong early expression like *Tc-zen1*. Yet, instead of a lack of function or shadow redundancy to Tc-Zen1, we uncover a distinct early role of Tc-Zen2 in the regulation of a key subset of genes, including the rapid repression of *Tc-zen1*. RNA-seq data also reveal subtle aspects of temporal variability (heterochrony) after *Tc-zen2* RNA interference (RNAi) that affect late morphogenesis. Our

validation of specific transcriptional targets opens new avenues into serosal tissue biology and identifies a novel, paralogue-based regulatory circuit at the developmental transition from specification to maturation of the serosa. This now raises the question of how species with a single *zen* gene compare for the precision and progression of EEM development, and whether their molecular phenotypes support early Tc-Zen2 function as the consequence of iterative subfunctionalization events.

## Results

**Recent tandem duplication of *zen* in the *Tribolium* lineage**. We first surveyed *Tribolium* beetle genomes to assess sequence conservation at the *Hox3* locus. Using the *T. castaneum* paralogues as BLASTn queries, we find that the tandem duplication of *zen* is conserved across three closely related congenerics: *Tribolium freemani*, *Tribolium madens*, and *Tribolium confusum* (Fig. 1a, 14–61 million years divergence[27]). Consistent with a recent event, phylogenetic analysis supports a single duplication at the base of the *Tribolium* lineage, and sequence alignments show particularly strong conservation in the homeobox, encoding the DNA-binding homeodomain (Fig. 1b; Supplementary Figs. 1 and 2).

Next, we investigated levels of coding sequence conservation between the *T. castaneum zen* (hereafter "*Tc-zen*") paralogues. Strongest nucleotide conservation occurs within the homeobox, where three conservation peaks correspond to the three encoded α-helices (Fig. 1c: >80% identity). In fact, within the coding sequence for the third α-helix there is a 20-bp stretch with 100% nucleotide identity (Fig. 1c), which is roughly the effective length of sequence for achieving systemic knockdown by RNAi[28]. Indeed, *Tc-zen1*-specific double-stranded RNA (dsRNA) that spans the homeobox is sufficient to effect significant cross-paralogue knockdown of *Tc-zen2* (Fig. 1d: 19% of wild-type expression with the long dsRNA fragment compared to only 35% with the short fragment; beta regression, odds ratio of the short fragment vs. long fragment $= 2.36$, 95% CI $= 1.65$–$3.37$, $p = 2.38e^{-6}$). As *Tc-zen2* is expressed in the tissue domain specified by Tc-Zen1[26], it is in fact surprising that any residual *Tc-zen2* transcript is detected in the *Tc-zen1* RNAi background, an aspect we revisit in the "Discussion". Meanwhile, for our experimental design we find that indeed a short fragment alone is sufficient to strongly knock down *Tc-zen1* itself (no significant change in knockdown efficiency between the long and short fragments: 10 and 11% expression, respectively; beta regression, odds ratio of the short fragment vs. long fragment $= 1.12$, 95% CI $= 0.75$–$1.66$, $p = 0.577$). For all subsequent functional testing we thus designed our dsRNA fragments to exclude the homeobox, thereby avoiding off-target effects and ensuring paralogue-specific knockdown (Fig. 1c: *Tc-zen1* short fragment: yellow; *Tc-zen2*: green; no ≥20-bp stretches of nucleotide identity in these regions).

**Distinct paralogue roles at different developmental stages**. EEM development has been well characterized morphologically in the beetle[13,29–31], including the *Tc-zen* paralogues' roles. Briefly, the first differentiation event distinguishes the serosa from the rest of the blastodermal cell sheet (Fig. 2a, at ~10% embryonic development). Tissue reorganization then involves serosal expansion and internalization of the embryo and amnion (EEM formation: subdivided into the "primitive pit" and "serosal window" stages). This topology is later reversed when the EEMs actively rupture and contract ("withdrawal"), coordinated with expansion of the embryo's flanks for dorsal closure of the body (Fig. 2c, at ~75% development). After *Tc-zen1* RNAi, presumptive serosal cells are respecified to other anterior fates, leading to an early enlargement of the head and amnion and loss of serosal identity (Fig. 2b)[26].

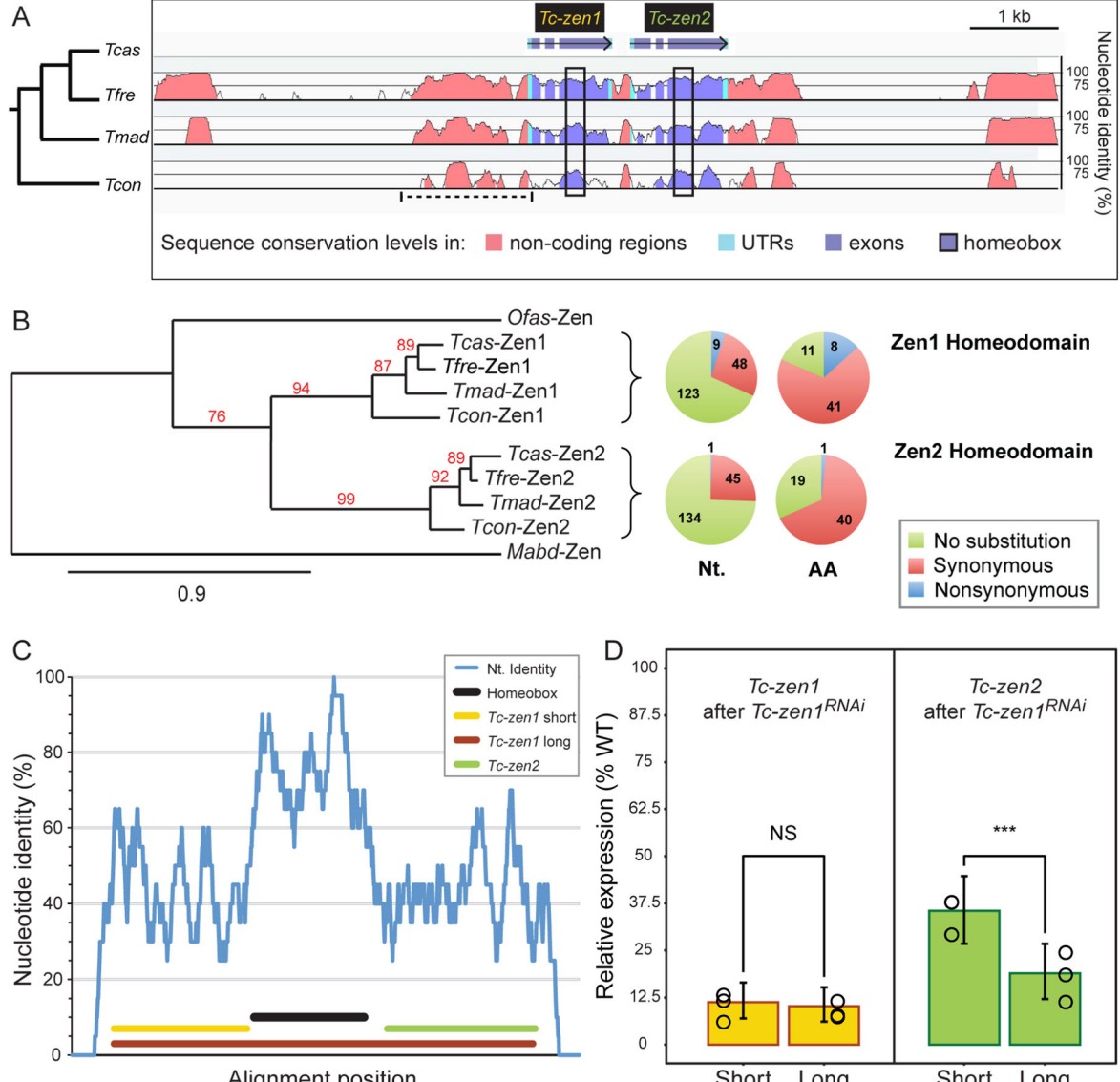

**Fig. 1 High sequence conservation of *Tribolium zen* orthologues and paralogues. a** Sequence conservation at the *Hox3/zen* locus of four *Tribolium* congenerics (represented by four-letter species abbreviations), using the *T. castaneum* locus as the reference sequence, assayed with a 100-bp sliding window. Non-coding sequence (pink) includes a highly conserved region that was recently tested as a *Tc-zen1* reporter[39] (dashed line, see "Discussion"). **b** Representative phylogeny supporting a lineage-specific clade for the *Tribolium* Zen proteins (see "Methods", Supplementary Fig. 2), with single-copy orthologue outgroups for the specification and morphogenesis functions (*Megaselia abdita*[18] and *Oncopeltus fasciatus*[17], respectively). Node support values (%) are indicated; branch length unit is substitutions per site. The pie charts summarize substitutions at the nucleotide (Nt.) and amino acid (AA) levels within the homeobox of *Tribolium zen* orthologues (see also Supplementary Fig. 1). **c** *T. castaneum zen* paralogue coding sequence comparison, based on a 20-bp sliding window. The final alignment of 887 positions includes 399 identities and 154 unaligned positions (gaps). Colored lines denote the homeobox and dsRNA fragments (see legend). **d** *Tc-zen1* RNAi with the long dsRNA fragment causes cross-paralogue knockdown, with significantly stronger reduction in *Tc-zen2* levels (beta regression tests: see main text). Mean expression levels (RT-qPCR) are shown from three biologically independent replicates after RNAi with the indicated dsRNA fragments (mapped in (**c**), same color code); error bars represent 95% confidence intervals; individual data points are shown, with the source data values given in Supplementary Data 5.

*Tc-zen2* RNAi impairs or wholly blocks late EEM withdrawal[12,26], confining the embryonic flanks such that the epidermis encloses the embryo's own legs instead of closing the back, leading to an everted (inside out) configuration (Fig. 2d)[12,32].

Here, we were able to reproduce the morphological phenotypes after RNAi for each *Tc-zen* paralogue (Fig. 2a′–d′). RNAi is particularly efficient for *Tc-zen1* (98.8% knockdown, Fig. 2e). Specific phenotypes after *Tc-zen2* RNAi (73.8% knockdown) include complete eversion (20.5%, Fig. 2d′) as well as milder defects in EEM withdrawal (53.3%, Fig. 2f; Supplementary Fig. 3). Furthermore, we newly explored how the paralogues' functions

relate to their transcript expression profiles across embryogenesis. Consistent with their functions, *Tc-zen1* has early expression while only *Tc-zen2* persists until the membrane rupture stage (Fig. 2g). Unexpectedly, late-acting *Tc-zen2* has peak transcript expression during early development.

## Subtle expression differences during early development. To gain insight into *Tc-zen* gene regulation and to determine the developmental stages of primary transcription factor function for each paralogue, we undertook a fine-scale spatiotemporal

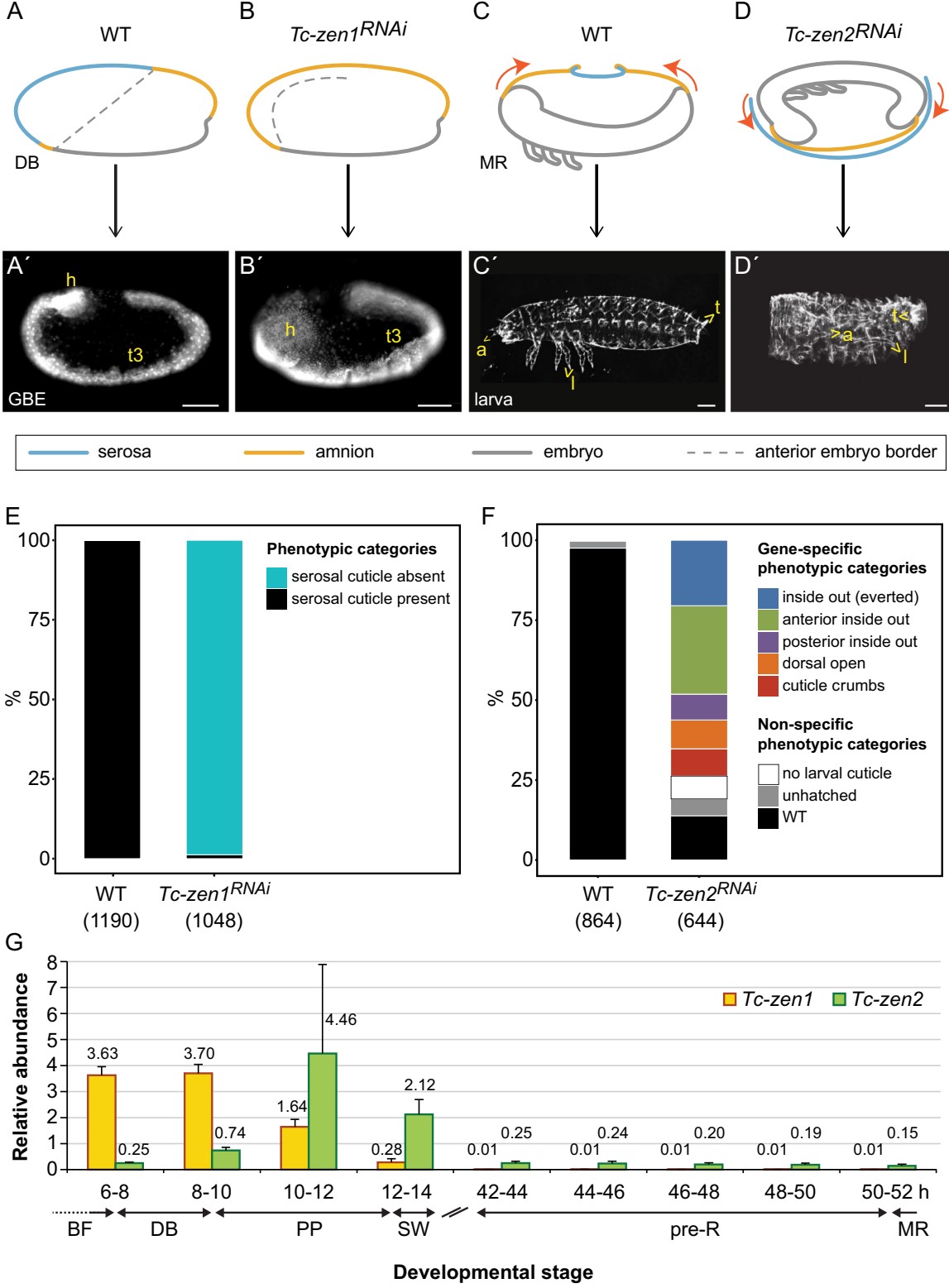

**Fig. 2 Tc-zen paralogue roles in early specification (Tc-zen1) or late EEM morphogenesis (Tc-zen2).** Phenotypic consequences of RNAi for the *Tc-zen* paralogues are depicted: schematically at the time of primary defect (**a–d**, upper row) and with micrographs for the resulting phenotypes (**a′–d′**, lower row: DAPI nuclear stain (**a′**, **b′**) and cuticle preparations (**c′**, **d′**); see also Supplementary Fig. 3). The dashed lines in the schematics in **a**, **b** denote the anterior embryonic border. Anatomical abbreviations: a antenna; h head; l leg; t telson; t3 third thoracic segment. Scale bars are 100 µm. **e**, **f** RNAi phenotypic penetrance per paralogue. Sample sizes are numbers of embryos. **g** Expression profiles of *Tc-zen1* and *Tc-zen2* in early (6–14 hAEL) and late (42–52 hAEL) development (RT-qPCR). Mean expression levels are shown from four biological replicates; error bars represent one standard deviation. For clarity, mean values are stated here, with the individual source data values given in Supplementary Data 5. Staging abbreviations: BF blastoderm formation/uniform blastoderm; DB differentiated blastoderm; PP primitive pit; SW serosal window; GBE extended germband; pre-R pre-rupture; MR extraembryonic membrane rupture. Time is hours after egg lay (hAEL).

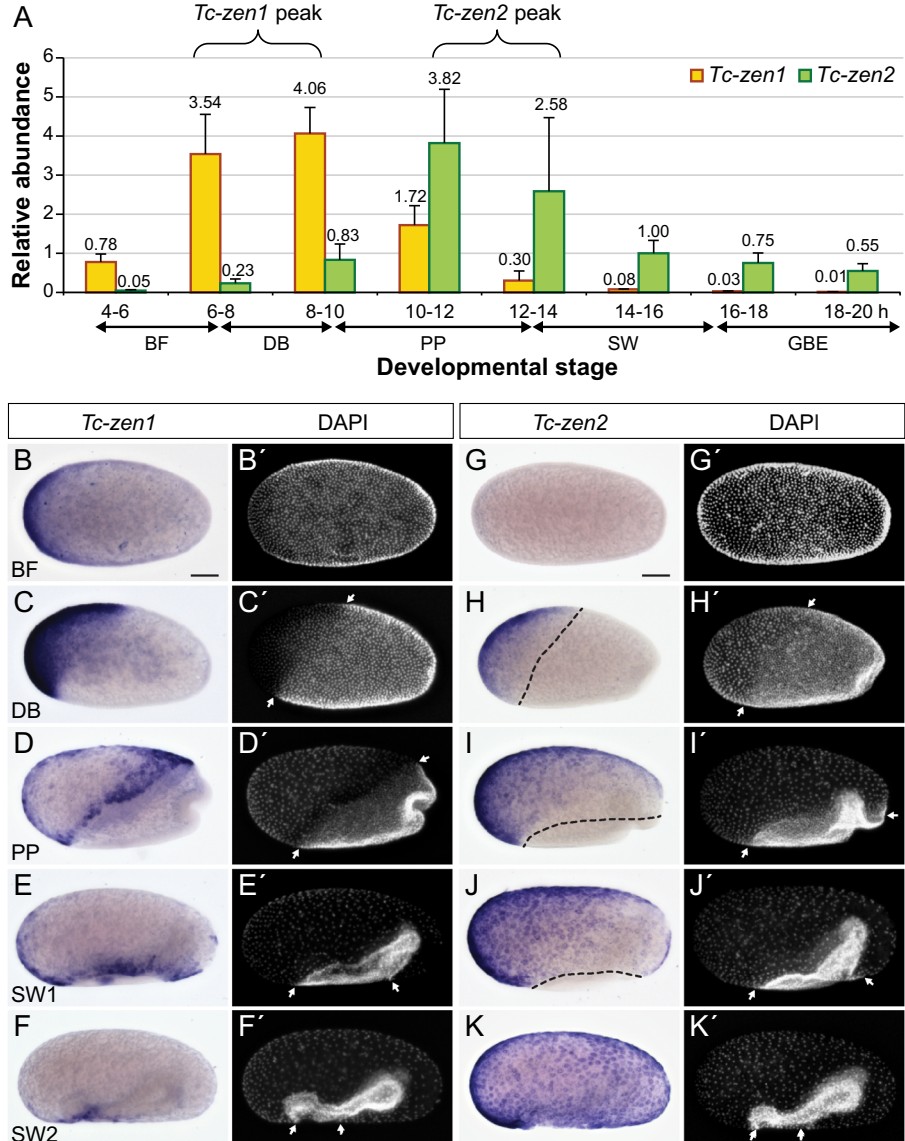

**Fig. 3 Transcript expression dynamics of the *Tc-zen* paralogues during early embryogenesis. a** Quantification of mean transcript levels in 2-h intervals (RT-qPCR), from four biological replicates; error bars represent one standard deviation. For clarity, mean values are stated here, with the individual source data values given in Supplementary Data 5. Whole mount in situ hybridization for *Tc-zen1* (**b–f**) and *Tc-zen2* (**g–k**), with nuclear counterstains for morphological staging (**b′–k′**, arrows label the expanding serosal border). All micrographs are oriented with anterior left and shown in lateral aspect with dorsal up (except in (**b**) and (**g**), which depict stages before this can be determined). Scale bars in (**b**) and (**g**) are 100 μm and apply to (**b–f′**) and (**g–k′**), respectively.

characterization of *Tc-zen1* and *Tc-zen2* expression for both transcript and protein (RT-qPCR, in situ hybridization, western blotting, immunohistochemistry).

As both paralogues are strongly expressed in early development (Fig. 2g), we first examined these stages in detail. *Tc-zen1* transcript arises in an anterior gradient during blastoderm formation (4–6 h after egg lay, hAEL), peaks at the differentiated blastoderm stage with uniform expression throughout the presumptive serosa (6–10 hAEL), and then becomes patchy and retracts to a narrow region at the tissue's border during EEM formation (10–14 hAEL; Fig. 3a–f). After the EEMs have fully enclosed the early embryo, *Tc-zen1* transcript is no longer detected (Figs. 2g and 3a). Peak *Tc-zen1* transcript expression is followed shortly by detectable protein for Tc-Zen1, although this, too, only occurs during early development (Fig. 4a; Supplementary Figs. 4 and 5).

*Tc-zen2* expression begins slightly later, at the differentiated blastoderm stage (6–8 hAEL), with peak levels occurring during EEM formation (10–14 hAEL; Fig. 3a). We also observed spatial differences between the paralogues. *Tc-zen2* is first detected only in an anterior subset of the serosa when *Tc-zen1* is expressed in the entire tissue (compare Fig. 3c with 3h). Then, *Tc-zen2* transcript expands throughout the serosa while *Tc-zen1* transcript retracts, concomitant with the expansion of the entire serosal tissue during EEM formation (compare Fig. 3d–f with 3i–k; Supplementary Fig. 5). Notably, the *Tc-zen* paralogues are expressed consecutively, but not concurrently, at the rim of the serosa. It is only during late EEM formation that we first observe *Tc-zen2* expression throughout the entire serosal tissue (Fig. 3k). By this time, Tc-Zen2 protein is also strongly expressed and persists (Fig. 4a, b; Supplementary Figs. 4, 5, and see below), while *Tc-zen2* transcript wanes gradually (from 14 hAEL; Fig. 3a).

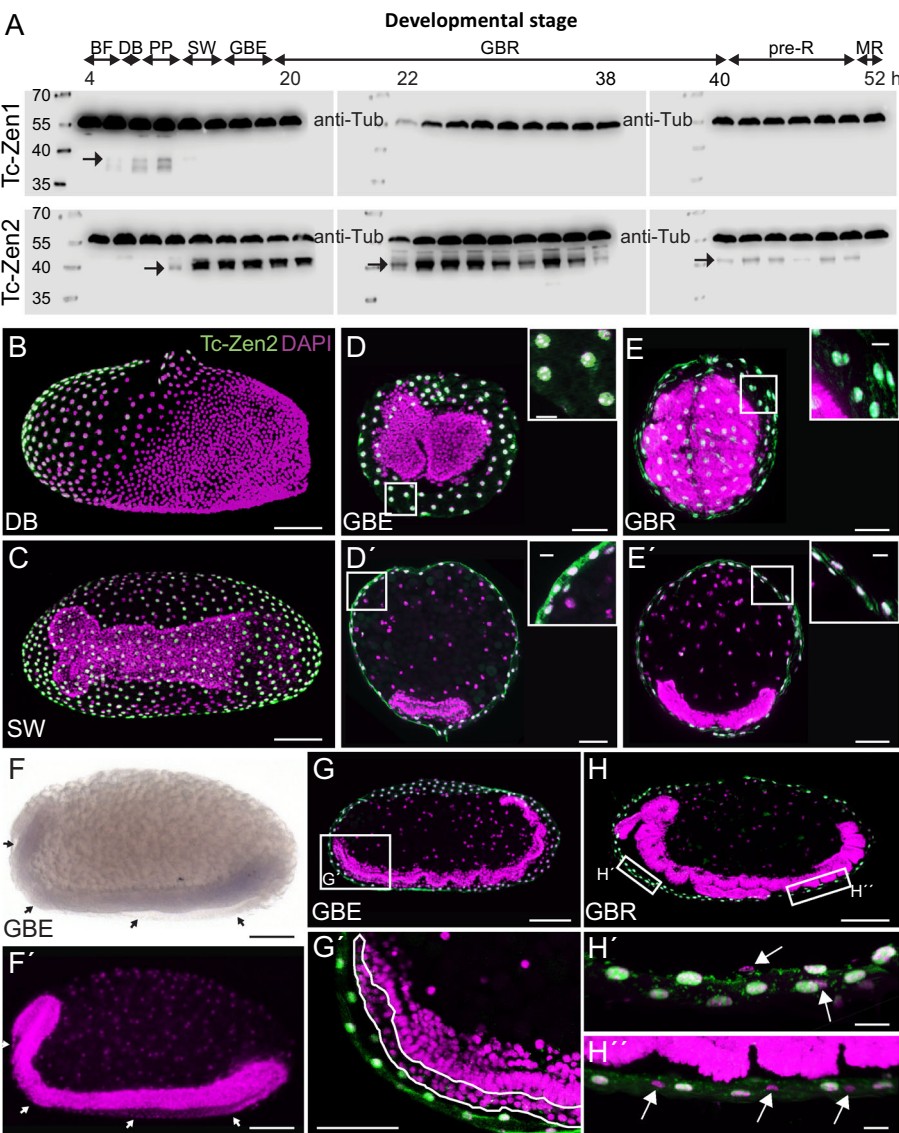

**Fig. 4 Paralogue-specific Tc-Zen protein expression time courses and spatial restriction of Tc-Zen2. a** Tc-Zen1 and Tc-Zen2 expression profiles, spanning 4–54 hAEL (western blots, with anti-Tubulin as an internal loading control at 59.0 kDa; left-hand size standards are in kDa). Each blot shows a chronological assay in 2-h intervals (labels indicate the minimum age from each interval). Tc-Zen1 protein (upper blots, arrow) is briefly detectable, from blastoderm differentiation until early serosal window closure. Tc-Zen2 (lower blots, arrows) arises at the primitive pit stage and persists until membrane rupture. See also Supplementary Fig. 4 for additional blots. Staging abbreviations as in Figs. 2 and 3, and: GBR germband retraction. Staining in whole mount (**b**), (**c**), (**f**) and cryosectioned (**d**), (**e**), (**g**), (**h**) preparations shows that Tc-Zen2 is specific to the serosa, with protein consistently localized to the nucleus. We found no evidence for Tc-zen2 expression in the second extraembryonic tissue, the amnion. Tc-Zen2-negative tissue and nuclei of the amnion are labeled with arrows (**f**, **f′**, **h′**, **h″**) or a dashed outline (**g′**). The specimen shown in (**f**), where the serosa was removed prior to staining, derives from a staining experiment spanning all early developmental stages, and where the serosa was specifically stained for Tc-zen2 transcript in other specimens, as seen in Fig. 3h–k. Images labeled with the same letter are of a single embryo. Nuclear counterstains are shown in magenta. Scale bars are 100 µm (whole mounts and overviews of longitudinal sections), 50 µm (transverse sections and (**g′**)), and 10 µm (insets and (**h′**, **h″**) focusing on nuclear localization).

**Early transcriptional impact of *Tc-zen1* and *Tc-zen2*.** Since protein expression follows shortly after peak transcript expression for both paralogues (Figs. 3a and 4a), we used the high sensitivity of our RT-qPCR survey (Fig. 3a) to inform our staging for functional testing by RNAi. To identify transcriptional targets for each *zen* gene, our RNA-seq after RNAi approach assessed differential expression (DE) between age-matched wild-type and knockdown samples. We focused specifically on the time windows of peak gene expression: 6–10 hAEL for *Tc-zen1* and 10–14 hAEL for *Tc-zen2* (curly brackets in Fig. 3a). These 4-h windows were chosen to maximize the number of identified target genes while prioritizing direct targets for Zen transcription factor binding.

The RNA-seq data are consistent with a priori expectations based on the morphological consequences of RNAi for each *zen* gene (Fig. 2a–d). That is, *Tc-zen1* has a clear early role in tissue specification, and its knockdown at these stages has a strong transcriptional impact, wherein principal component analysis (PCA) clearly distinguishes experimental treatments (Fig. 5a). In contrast, *Tc-zen2* has an early expression peak but its manifest role in late EEM withdrawal occurs nearly 2 days later (56% development later). Accordingly, we find a negligible effect on the early egg's total transcriptome after *Tc-zen2* RNAi (Fig. 5a), despite verification of efficient knockdown (Fig. 2f). RNAi efficiency was also confirmed directly with the RNA-seq data:

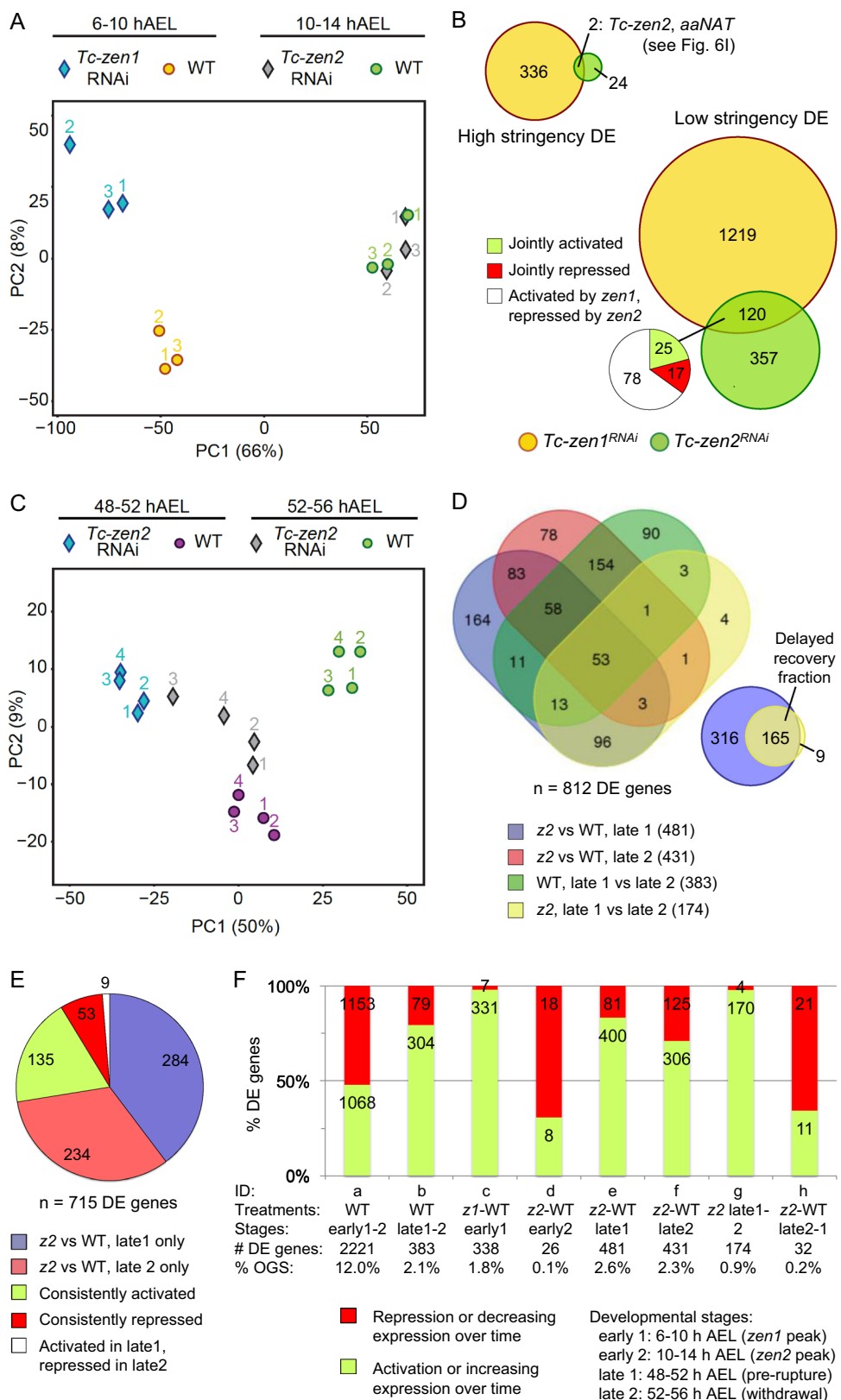

both *Tc-zen1* and *Tc-zen2* exhibit DE reduction after their respective knockdown (Supplementary Data 1A, B: significant fold change reductions of $-4.86$ for *Tc-zen1* and $-3.54$ for *Tc-zen2*). Overall, we obtained 338 DE genes after *Tc-zen1* RNAi compared to only 26 DE genes after *Tc-zen2* RNAi, while global transcriptional changes affect nearly 12% of all genes during early

embryogenesis (2221 DE genes; threshold of $P_{adj} \leq 0.01$ and $|FC| \geq 2$ for all DE genes: Fig. 5f bar chart elements a, c, d; Supplementary Data 1A–C).

Given the recent nature of the duplication, which is evident in the similarity of the *Tc-zen* paralogues' DNA-binding homeodomains and early expression profiles, we asked whether there is

**Fig. 5 *Tc-zen* genes' transcriptional control: relative impact, overlap, and stage-specific dynamics revealed by RNA-seq after RNAi. a** PCA score plot for *Tc-zen1* and *Tc-zen2* and their respective age-matched wild-type samples in early development, at the stages of peak expression (three biological replicates). **b** Venn diagram assessments of shared early targets of the *Tc-zen* paralogues, with high ($P_{adj} \leq 0.01$, $|FC| \geq 2$) or low ($P_{adj} \leq 0.05$, $|FC| > 1$) stringency DE criteria. Subset pie chart summarizes the direction of regulation for shared targets (see legend). *Tc-zen2* appears as a shared high-stringency target, reflecting its direct experimental knockdown by *Tc-zen2* RNAi as well as Tc-Zen1's role in its endogenous upregulation (see below). **c** PCA score plot of *Tc-zen2^{RNAi}* and wild-type samples in late development, at the stages before and during EEM withdrawal (four biological replicates). **d** Venn diagrams of numbers of late DE genes across pairwise comparisons (see legend, with total DE gene counts per comparison listed parenthetically; see also **f** bar chart elements b, e, f, g). **e** Detailed comparisons of DE genes by stage (pre-rupture, withdrawal) and direction of regulation (activation, inhibition). Values are the number of DE genes. See also **f** bar chart elements e, f. **f** Summary metrics indicate the number of DE genes per pairwise comparison, and the proportion that are activated or inhibited by a given *Tc-zen* paralogue or that increase or decrease in expression over time in a given background (WT or RNAi). DE cut-off values are $P_{adj} \leq 0.01$, $|FC| \geq 2$. For complete gene lists and statistics, see Supplementary Data 1A–C and 3A–E. Treatment abbreviations: WT wild type; *z1 Tc-zen1^{RNAi}*; *z2 Tc-zen2^{RNAi}*. Staging for all panels is summarized in the lower right legend.

a legacy of shared early function. If this is the case, *Tc-zen2* might exhibit a subtle regulatory profile similar to *Tc-zen1*. However, even with relaxed thresholds for DE, we find few shared targets between the paralogues, particularly when the direction of regulation is considered (Fig. 5b; Supplementary Data 2A, B). Thus, we conclude that *Tc-zen2* has a minimal effect on early development, and that this does not constitute a transcriptional "echo" of co-regulation with *Tc-zen1* due to common ancestry. Why, then, is *Tc-zen2* strongly expressed during early development?

**The *Tc-zen* paralogues are mutual regulatory targets**. We next considered the *Tc-zen* paralogues as factors necessary for defining the serosal tissue, as indicated by their specific transcriptional targets. *Tc-zen1* is strictly required for serosal tissue identity[26]. Differentiation of the serosa involves an early switch from mitosis to the endocycle[29,30], resulting in polyploidy[13]. Consistent with this, we identified a homolog of the endocycle factor *fizzy-related*[33,34] among DE genes upregulated by Tc-Zen1 (Supplementary Data 1A). From known targets of Tc-Zen1, we also recovered *Dorsocross* and *hindsight*, involved in EEM formation[35], and *chitin synthase 1*, required for production of the protective cuticle[10]. In addition, we hypothesized that the slight offset whereby *Tc-zen1* expression precedes *Tc-zen2* is consistent with Tc-Zen1 activating *Tc-zen2*. We could confirm this regulatory interaction both by RNA-seq and RT-qPCR after *Tc-zen1* RNAi (Fig. 6a, b). Thus, Tc-Zen1 as a serosal specifier upregulates factors for definitive tissue differentiation, including *Tc-zen2* as a candidate (Fig. 6i).

Are there Tc-Zen2 transcriptional targets that could support an early role in the serosa for this paralogue? Among the few genes with strong DE (Fig. 5f bar chart element d), we validated several as probable targets. These genes are expressed in the early serosa and/or their transcript levels are first strongly upregulated during peak *Tc-zen2* expression (12–14 hAEL; e.g., Supplementary Fig. 6). Their putative functions as enzymes or structural components for chitin-based cuticle (Cpr proteins) and as signaling molecules support a role for Tc-Zen2 in the physiological maturation of the serosa, providing complementary regulatory control to Tc-Zen1.

In performing reciprocal validation assays, we then uncovered an unexpected early function of Tc-Zen2 in the repression of its own paralogous activator. After *Tc-zen2* RNAi, *Tc-zen1* transcript is still detectable, a qualitative observation that previously had been taken as evidence for a lack of paralogue feedback[26]. The sensitivity of our quantification and spatial analyses across the dynamic early stages of development now allows us to correct this. We detect an upregulation of *Tc-zen1* that was only weakly suggested by our RNA-seq data but then strongly supported in RT-qPCR assays (Fig. 6a, b). We also confirmed this observation by in situ hybridization. After

*Tc-zen2* RNAi, *Tc-zen1* transcript is expressed at higher levels than in wild type (compare Fig. 6c, d with 6f, g). *Tc-zen1* also remains strongly expressed throughout the serosa at stages when its wild-type expression is restricted to low levels at the tissue rim (compare Fig. 6e with 6h). In fact, the abrupt reduction in *Tc-zen1* transcript and protein levels in wild type correlates with increasing *Tc-zen2* levels, and spatially their dynamic expression is largely complementary, if not outright mutually exclusive (Figs. 3 and 4). Together, these results suggest that Tc-Zen1 upregulates *Tc-zen2* in its wake, and that in turn early Tc-Zen2 represses *Tc-zen1*. Thus, the *Tribolium* paralogues function as mutual regulatory targets, comprising an integrated regulatory module for early serosal development (Fig. 6i).

**Tc-Zen2 is exclusively serosal and persistently nuclear**. To complete our analysis of *Tc-zen2*, we also examined its activity at later stages. We could detect both transcript (weakly, Figs. 2g and 3a) and protein (particularly strongly in mid-embryogenesis, Fig. 4a) continuously until the stage of EEM withdrawal, spanning 14–75% of development (10–54 hAEL, assayed in 2-h intervals; see also Supplementary Fig. 4). Moreover, we find that Tc-Zen2 is persistently localized to the nucleus, demonstrated by fluorescent immunohistochemistry on cryosectioned material of selected stages (Fig. 4b–e, g, h). This contrasts with some species' orthologues, which show stage-specific exclusion of Zen protein from the nucleus[36]. We could also refine the spatial scope of *Tc-zen2* activity: in contrast to earlier reports[26], we found no evidence for *Tc-zen2* transcript or protein in the amnion (Fig. 4d–h''), indicating that this factor is strictly serosal.

**Late transcriptional dynamics are largely *Tc-zen2*-dependent**. The early RNA-seq after RNAi experiment examined the time of peak *Tc-zen2* expression. Complementing this, we used the same approach to examine the stage of known *Tc-zen2* function in late EEM withdrawal. Withdrawal begins with rupture of the EEMs, at 52.1 ± 2.3 hAEL as determined by live imaging[31]. Here, we assayed the 4-h intervals just before (48–52 hAEL) and after (52–56 hAEL) rupture, to assess *Tc-zen2* transcriptional regulation that precedes and then accompanies withdrawal. Consistent with *Tc-zen2*'s known role, we detect >16× more DE genes after *Tc-zen2* RNAi in late development (>430 DE genes, compare Fig. 5f bar chart elements e, f with 5f bar chart element d). PCA also clearly separates knockdown and wild-type samples at late stages (Fig. 5c).

Our staging helps to contextualize *Tc-zen2* and EEM-specific processes relative to concurrent embryonic development. We thus evaluated DE in pairwise comparisons not only between wild-type and RNAi samples, but also over time in both backgrounds (Fig. 5d, f bar chart elements b, e–g). Comparisons across

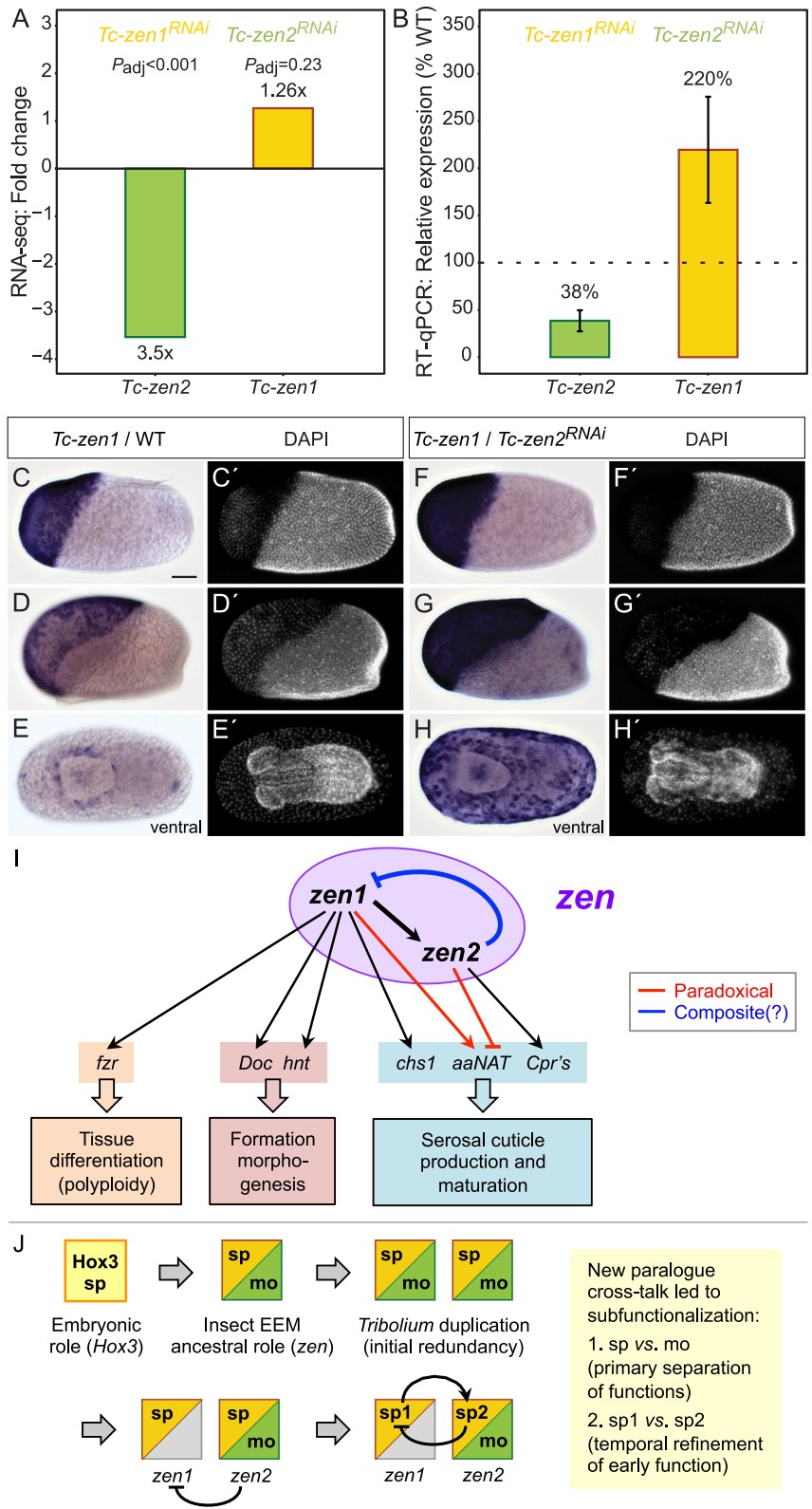

consecutive stages in early and late development (Fig. 5f bar chart elements a, b) reveal two general changes in the wild-type transcriptional landscape. There is far less dynamic change in gene expression in late development (5.8× fewer DE genes), consistent with steady state and ongoing processes in later embryogenesis compared to the rapid changes of early development. Also, whereas early development shows a fairly even

balance between up- (48%) and downregulation (52%), late development is predominantly characterized by increasing expression levels over time (79%).

Against this backdrop, the transcriptional impact of *Tc-zen2* is quite pronounced. Most genes with changing expression over time in the late wild-type background are also affected by *Tc-zen2* RNAi (Fig. 5d: 77%, 293/383 DE genes from green Venn diagram

**Fig. 6 Evidence and implications of a *zen* paralogue regulatory module. a, b** *Tc-zen1* and *Tc-zen2* as candidate transcriptional targets of the respective paralogue, determined in comparisons of wild-type and RNAi samples at the stages of peak paralogue expression (see Fig. 3a). Regulatory changes are corroborated by assays from RNA-seq (**a**: fold change, $P_{adj}$, as given from three biological replicates in Supplementary Data 1A, B) and RT-qPCR (**b**: mean relative expression, error bars represent one standard deviation, from biologically independent samples $n = 60$ for *Tc-zen2* expression and $n = 33$ for *Tc-zen1* expression, as given in Supplementary Data 5). Whole mount in situ hybridization for *Tc-zen1* in wild type (**c–e**) and after *Tc-zen2* RNAi (**f–h**), with nuclear counterstains for morphological staging (**c′–h′**). Wild-type and RNAi eggs were stained for the same duration. All micrographs are oriented with anterior left and shown in lateral aspect with dorsal up, unless stated otherwise. Scale bar in (**c**) is 100 μm and applies to all micrographs. **i** The *Tc-zen* paralogues comprise an integrated "zen" regulatory module for early serosal development. Paradoxical regulation (red) involves competing inputs to the same target. Composite regulation (blue) denotes potential repression at both the transcriptional and translational levels. Gene abbreviations correspond to the full names and descriptions in the main text. **j** Evolutionary scenario for the ancestral extraembryonic functions of Zen and its subsequent changes in *Tribolium*. Progressive partitioning of the functions occurred through stepwise acquisition of paralogue mutual regulation, resulting in iterative subfunctionalization. Primary subfunctionalization separated early specification (sp) and late morphogenesis (mo) functions. Secondly, the original specification function has now been finely subdivided into the initial, substantial role that Tc-Zen1 continues to play ("sp1") and the slightly later, subtle functions of Tc-Zen2 that we reveal here ("sp2").

set). We detect this strong effect even though *Tc-zen2* is restricted to the serosa (Fig. 4), a tissue that ceased mitosis (Fig. 6i) and comprises only a small cell population within our whole-egg samples. This suggests that most dynamic transcription in late development pertains to EEM morphogenesis, with the global transcriptional impact of *Tc-zen2* at these stages even greater than for *Tc-zen1* in early development (compare Fig. 5f bar chart elements e, f with 5f bar chart element c). Most candidate *Tc-zen2* targets are differentially expressed at a single stage (72%), although a substantial fraction (26%) exhibits consistent activation or repression, while an intriguing handful of genes shows changing, stage-specific regulation (Fig. 5e). These patterns imply that the persistent nuclear localization of Tc-Zen2 (Fig. 4) reflects active transcriptional control, not merely localization to the nucleus or DNA binding in a paused, nonfunctional state[37].

To characterize late Tc-Zen2 activity, we functionally annotated and validated candidate transcriptional targets. Gene ontology (GO) enrichment tests confirmed that ongoing cuticle regulation is a primary role, including remodeling as the serosa detaches from its own cuticle in preparation for withdrawal (Supplementary Fig. 7, Supplementary Data 4A, B). For validation, we selected a dozen genes based on known biological processes for tissue remodeling (e.g., cytoskeleton and morphogenesis), prominent GO categories (e.g., transmembrane transporters), and evidence of dynamic regulation (Fig. 5e; Supplementary Fig. 8, Supplementary Tables 1 and 2). All tested candidates were confirmed by RT-qPCR (Supplementary Fig. 8). This included two of the genes that are first activated and then repressed by Tc-Zen2, where both genes encode proteins with conserved domains of unknown function (Supplementary Table 2). Notably, such dynamic targets include a subset of the *Osiris* multigene locus (Supplementary Fig. 8)[38]. Lastly, we evaluated Tc-Zen2 regulation of serosal immune genes[7]. Although our samples were not pathogen challenged, we could detect expression for 83% of these genes (89 of 107 genes), with 20% showing DE after *Tc-zen2* RNAi (Supplementary Data 3A, B). Thus, while Tc-Zen2 is not a global effector, it may regulate subsets of immune genes. Notably, transcripts of most serosal immune genes (87 genes) continue to be detected during withdrawal, supporting their expression as an inherent feature of the serosa—even when it is no longer a protective layer enclosing the embryo.

**Evidence of variable developmental delay after *Tc-zen2* RNAi.** The *Tc-zen2^{RNAi}* molecular phenotype also provides new insight into the physical phenotype of defective EEM withdrawal, suggesting that a variable, partial delay in preparatory transcriptional changes is the underlying cause.

Several observations are consistent with a delay. As noted above, all late RNA-seq biological replicates cluster by treatment in PCA. Interestingly, the older *Tc-zen2^{RNAi}* samples (52–56 hAEL) have intermediate component scores compared to the younger *Tc-zen2^{RNAi}* and younger wild-type samples (48–52 hAEL, Fig. 5c). Similarly, DE comparisons identify noticeably fewer DE genes between the older *Tc-zen2^{RNAi}* sample and either of the younger samples (Fig. 5f bar chart elements g, h; Supplementary Data 3D, E). In fact, the very low number of DE genes implies that there is virtually no difference in the transcriptional profile of the older *Tc-zen2^{RNAi}* sample compared to the younger wild-type sample (Fig. 5f bar chart element h). Furthermore, nearly all genes that change in expression over time in the *Tc-zen2^{RNAi}* background are also candidate targets of Tc-Zen2 at the prerupture stage (95%, Fig. 5d: inset Venn diagram). In other words, *Tc-zen2^{RNAi}* eggs generally require an additional 4 h (5.6% development) to attain a transcriptional profile comparable to the wild-type pre-rupture stage, and this is achieved by belated activation of Tc-Zen2 target genes. However, only a subset of genes exhibit delayed recovery (34%, Fig. 5d inset). These target genes may thus be independently activated by other factors, in addition to activation by Tc-Zen2.

Our RNA-seq data also reveal increased variability after *Tc-zen2* RNAi. The pre-rupture *Tc-zen2^{RNAi}* biological replicates show comparably tight clustering to their age-matched wild-type counterparts (48–52 hAEL, Fig. 5c). This suggests that pre-rupture is the stage of primary Tc-Zen2 function, also supported by our detection of the greatest number of DE genes at this stage (compare Fig. 5f bar chart elements d, e, f). In contrast, the older *Tc-zen2^{RNAi}* samples have a noticeably greater spread along the vectors of the first two principal components (52–56 hAEL, Fig. 5c), consistent with cumulative variability as the RNAi phenotype develops, presumably in part due to the observed partial transcriptional recovery (Fig. 5d). This variability may in itself provide explanatory power for the spectrum of end-stage *Tc-zen2^{RNAi}* phenotypes (Fig. 2f, see below).

## Discussion

Our analysis of regulation upstream and downstream of the beetle *zen* genes reveals several unexpected features regarding the evolution and biological roles of these unusual paralogues.

First, sequence conservation belies the extent of *zen* paralogue functional divergence. Fine-tuned transcriptional regulation is required to restrict regulatory crosstalk, and conserved noncoding regions may contribute to this. The region upstream of *zen1* has particularly high conservation and was tested as an in vivo *Tc-zen1* reporter in a recent study[39] (Fig. 1a: dashed line region). This construct recapitulates expression at the rim of the serosal window, a feature common to both paralogues (as in

Figs. 3f, k and 6e). However, early blastoderm *Tc-zen1* expression is absent (cf., Fig. 3b, c), while subsequent embryonic/amniotic expression of this construct represents a wholly ectopic domain[39]. Our own observations that detectable, residual levels of *Tc-zen2* persist after *Tc-zen1* RNAi (Fig. 1d: short fragment) and that low but detectable levels of *Tc-zen2* are expressed during blastoderm formation (4–6 hAEL, compare Fig. 3a with 3g) imply early *Tc-zen2* regulation that is independent of Tc-Zen1, possibly involving unknown anterior terminal activators relevant for *Tc-zen1* itself. Thus, regulation of the *Tc-zen* genes requires multiple inputs that remain to be elucidated.

Specificity of regulation by the *Tc-zen* genes is also elusive. In the *Tc-zen* homeobox, sequence similarity is particularly high in the third α-helix, which confers DNA-binding specificity (Fig. 1c, d)[20,40,41]. Yet, the paralogues' shared ancestry is not reflected in redundant activity (Fig. 5a, b). Rather, strong conservation across *Tribolium* species, particularly of *zen2* (Supplementary Fig. 1), may indicate not only limited divergence but also positive, purifying selection[42]. How, then, do the paralogues regulate different targets? In canonical Hox3 proteins, DNA-binding specificity can be enhanced by the common Hox co-factor Extradenticle[41]. In contrast, insect Zen proteins have lost the hexapeptide motif required for this interaction, and no other co-factor binding motifs are known[16], deepening the long recognized "Hox specificity paradox"[43] in the case of the beetle *zen* paralogues.

Recent work has revealed a role for differential chromatin accessibility in conferring Hox binding specificity[44]. This highlights an intriguing direction for future research, on the extent of rapid functional genomic changes that may occur in early development. However, the stage-specific and gene-specific regulatory activity of Tc-Zen2 (Fig. 5e; Supplementary Fig. 8) argues for greater regulatory precision than expected based on opportunistic binding to accessible targets alone.

Meanwhile, our molecular dissection of the *Tc-zen* paralogues elucidates their functional divergence. Specifically, mutual regulation has implications for the paralogues' network logic and confers temporal precision. The newly discovered negative feedback loop of Tc-Zen1 activation leading to repression by Tc-Zen2 constitutes a tight linkage. To what extent could *Tc-zen1* overexpression bypass upregulation of *Tc-zen2* as its target, resulting in repression of *Tc-zen1* and thus canceling out the manipulation? In fact *Tc-zen2* RNAi does confer overexpression of *Tc-zen1* and reduced *Tc-zen2* (Fig. 6; Supplementary Data 1B). However, consistently lower knockdown efficiency for *Tc-zen2* than for *Tc-zen1* (Fig. 2, ref. [26]) may reflect a dose-limiting lack of regulatory disentanglement. Arguably, *Tc-zen1* and *Tc-zen2* together satisfy the criteria of a minimal gene regulatory network (GRN) kernel[45], including "recursive wiring" and the experimental challenges this entails. Alternatively, the *Tc-zen* paralogues could be viewed as a single unit in a serosal GRN and thus qualify as a "paradoxical component" that both activates and inhibits (Fig. 6i)[46]. Consistent with theoretical expectations, delayed inhibition produces a discrete pulse of *Tc-zen1* (Figs. 3 and 4). As the pulse is non-oscillatory, this may also imply that *Tc-zen2* is a positive autoregulator[46], a property known for a number of canonical Hox genes[47,48] but thus far not known for insect *zen* genes.

Furthermore, Tc-Zen2 was previously implicated in the unusual role of translational repression of the early embryonic factor *caudal*[49]. Conceivably, Tc-Zen2 repression of *Tc-zen1* could act in a composite fashion, at both the transcriptional and translational levels. Composite activity could expedite repression—consistent with *Tc-zen1*'s abrupt decline (Figs. 3a and 4a)—and enhance stability of the system[50]. Similar regulatory dynamics have also been found in other contexts. Feedback loops with activation leading to inhibition can promote robustness[51] and rapidly, precisely restrict expression[52,53]. Whereas spatial precision

contributes to patterning of distinct tissues[53], the serosal Zen feedback loop generates temporal precision.

In other words, paralogue divergence promotes the progression of serosal development. Negative feedback implies a strong developmental requirement to repress *Tc-zen1*, even before the serosa has fully enclosed the embryo. Since *Tc-zen2* persists in this same domain, why is this necessary? In *Drosophila*, expression of *Dm-zen* is also short-lived[54], and its overexpression causes an increase in amnioserosal cell and nuclear size[55]. The insect EEMs are known to be polyploid to characteristic levels[13,56,57], and excessive ploidy could interfere with the tissues' structure and function as barrier epithelia[58]. Our RNA-seq DE analyses support a role for *Tc-zen1*, but not *Tc-zen2*, in promoting serosal endoreplication (Fig. 6i). Thus Tc-Zen2's repression of *Tc-zen1* may ensure a limited time window for this transition. The temporal offset also limits the amount of gene product of the paralogues' sole shared target that we identified, the cuticle maturation factor *aaNAT*, and overall effects temporally graded cuticle production (Figs. 5b and 6i; Supplementary Fig. 6). Together, the distinct roles of the *Tc-zen* paralogues offer an evolutionarily novel opportunity for regulatory refinement in the early insect serosa, with a finely tuned genetic separation of specification and maturation functions that fosters developmental progression.

Stepping back to a macroevolutionary framework, what are the implications of the *Tc-zen* paralogues for the evolution of insect *zen*? Although a specification function has only been demonstrated in the Holometabola[17–19,26,55], early expression is also known from some hemimetabolous species[36,59]. This suggests that the ancestral *zen* may have fulfilled both early specification and late morphogenesis roles. Then, the prominent functions of the *Tc-zen* paralogues would represent an instance of subfunctionalization[60]. Furthermore, *Tc-zen1* and *Dm-zen* differ from all other known homologs in lacking persistent serosal expression. Although the implications of *Dm-zen* temporal restriction have been extensively discussed[54,61], its downregulation is likely passive[62,63] and occurs much later than for *Tc-zen1*. Meanwhile, the *Tribolium* innovation of having two functional, early extraembryonic copies of *zen* may have originated as redundant early expression, but Tc-Zen2's repression of *Tc-zen1* constitutes a new regulatory link that ensures Tc-Zen2 is the sole functional *zen* gene for late morphogenesis. This repression thus subdivides a composite ancestral role between the paralogues: *Tc-zen1* for early specification, *Tc-zen2* for late morphogenesis. A second regulatory innovation, Tc-Zen1's activation of *Tc-zen2*, then conferred the temporal offset in early paralogue activity and resulted in a second, nested instance of subfunctionalization, for serosal specification (Fig. 6i, j). Thus, whereas the loss of autoregulation may have contributed to subfunctionalization of certain Hox genes among the four paralogous Hox clusters of vertebrates[64], here we infer that the acquisition of paralogue mutual regulation drove active, iterative subfunctionalization.

We have uncovered multiple roles of *Tc-zen2* as a diverged Hox gene throughout the lifetime of the serosal tissue, itself a morphological innovation[5]. Early, Tc-Zen2's repression of *Tc-zen1* (Fig. 6) and *Tc-caudal*[49] is noteworthy. A predominantly repressive role contrasts with Hox genes typically serving as activators, as do both *Tc-zen* paralogues at the stages of their primary function (Fig. 5f bar chart elements c, e). Also, the precise mechanism and targets of potential Tc-Zen2 translational repression remain open questions. Future work will clarify whether such a function arose independently in *Tribolium* Zen2 and dipteran Bicoid[20,23,40] as distinct *Hox3/zen* derivatives. In later development, the serosa is the cellular interface with the outer environment. Our data elucidate Tc-Zen2's roles in the known protective functions of cuticle formation[10] and innate immunity[7].

Beyond this, our DE gene sets comprise a large, unbiased sample of candidate targets, laying the foundation for investigating wider roles of *Tc-zen2* in this critical tissue.

Finally, we identify *Tc-zen2*-dependent EEM withdrawal as the major transcriptionally regulated event in late embryogenesis and assess its precision (Fig. 5d). Temporal and molecular variability after *Tc-zen2* RNAi underpins observed variability in knockdown EEM tissue structure, integrity, and morphogenetic competence, defining the broad spectrum of end-stage phenotypes (Fig. 2; Supplementary Fig. 3). This ranges from mild defects in dorsal closure after transient EEM obstruction to persistently closed EEMs that cause complete eversion of the embryo (Fig. 2; Supplementary Fig. 3, ref. [12]). The unifying feature is a heterochronic shift of extraembryonic compared to embryonic developmental processes (delayed EEM withdrawal compared to epidermal outgrowth for dorsal closure).

There may also be species-specific differences in the timing of Zen function for withdrawal morphogenesis. The sole *zen* orthologue in the milkweed bug *Oncopeltus fasciatus* has a similarly persistent expression profile and specific role in withdrawal, termed "katatrepsis" in this and other hemimetabolous insects[17]. We previously observed a number of *Of-zen*-dependent, long-term morphological changes prior to rupture[65], contrasting with the more proximate effect of *Tc-zen2* (discussed above). Taking the work forward, it will be interesting to compare *Tc-zen2* and *Of-zen* transcriptional targets. Evaluating conserved regulatory features of EEM withdrawal across the breadth of the insects will clarify macroevolutionary patterns of change in the very process of epithelial morphogenesis.

## Methods

***Tribolium castaneum* stock husbandry**. All experiments were conducted with the San Bernardino wild-type strain, maintained under standard culturing conditions at 30 °C and 40–60% relative humidity[66].

**In silico analyses**. Draft genome assemblies for *T. freemani*, *T. madens*, and *T. confusum* were obtained as assembled scaffolds in FASTA format (version 26 March 2013 for each species), accessed from the BeetleBase.org FTP site at Kansas State University (ftp://ftp.bioinformatics.ksu.edu/pub/BeetleBase/). Transcripts for *Tc-zen1* (TC000921-RA) and *Tc-zen2* (TC000922-RA) were obtained from the *T. castaneum* official gene set 3 (OGS3)[67]. These sequences were used as queries for BLASTn searches in the other species' genomes (BLAST + 2.2.30)[68,69]. Sequences were extracted to comprise the *Hox3/zen* genomic loci, spanning the interval from 5 kb upstream of the BLASTn hit for the 5′ UTR of *Tc-zen1* to 5 kb downstream of the BLASTn hit for the 3′ UTR of *Tc-zen2*. These genomic loci were then aligned with the mVista tool[70,71] using default parameters. Nucleotide identities were calculated for a sliding window of 100 bp.

The maximum likelihood phylogenetic tree (Fig. 1b) was constructed based on an alignment of full-length Zen proteins, with gaps permitted, using the Phylogeny.fr default pipeline settings, with MUSCLE v3.8.31 alignment and PhyML v3.1 phylogenetic reconstruction[72]. The same topology and comparable support values were also obtained with additional sequences and other methods (Supplementary Fig. 2). This includes *Drosophila* Zen and/or Z2 and/or *Drosophila* and *Megaselia* Bicoid, and/or insect Zen proteins with known expression but uncharacterized function[36,59], and bioinformatic predictions of Zen proteins in newly sequenced genomes. This also holds for trees generated with Bayesian methods from the same interface (MrBayes program v3.2.6, 1000 generations, 100 burn-in trees). That is, the *Tribolium* Zen proteins form a clear clade with various Zen outgroups, with the fly proteins as long branch outgroups to other Zen proteins.

Coding sequence for the *Tc-zen* paralogues was aligned with ClustalW[73], with manual curation to ensure a gap-free alignment of the homeobox. Nucleotide identities were calculated for a sliding window of 20 bp, using Simple Plot[74].

**RT-qPCR**. RNA was extracted using TRIzol Reagent (Ambion) according to the manufacturer's protocol. RNA quality was assessed by spectrophotometry (NanoDrop 2000, Thermo Fisher Scientific). cDNA was synthesized using the SuperScript VILO cDNA Synthesis Kit (Invitrogen). RT-qPCR was performed as described[35], using SYBR Green Master Mix (Life Technologies) and GoTaq qPCR Master Mix (Promega), with *Tc-RpS3* as the reference gene. Note that for *Tc-zen2* more consistent results were obtained using SYBR Green Master Mix. "Relative abundance" was calculated for each sample as the ratio relative to a pooled template control with cDNA from all depicted samples (method as in ref. [35]). Samples were measured for the *Tc-zen* paralogues' wild-type expression profiles (four

biological replicates: Figs. 2g and 3a) and evaluation of knockdown strength (three biological replicates: Figs. 1d and 6b; Supplementary Fig. 8). Intron-spanning primers were used for each *Tc-zen* paralogue and the selected candidate target genes (Supplementary Table 3). RT-qPCR data were evaluated using LinRegPCR v12.16[75,76].

**Parental RNAi and knockdown assessments**. Parental RNAi was performed as described[26], with dsRNA synthesized with specific primers (Supplementary Table 3) and resuspended in double-distilled water (ddH$_2$O). Generally, 0.3–0.4 μg of dsRNA was used to inject one pupa.

Analysis of knockdown efficiency with different *Tc-zen1* dsRNA fragments involved statistical tests on RT-qPCR data. The strength of the *Tc-zen* paralogues' knockdown using short and long *Tc-zen1* dsRNA fragments (Fig. 1c, d) was tested with a beta regression analysis with logit link function in R v3.3.2[77] using the package betareg v3.1-0[78]. Expression of the *Tc-zen* paralogues in knockdown samples relative to wild type was used as the response variable and dsRNA fragment length as the explanatory variable.

For *Tc-zen1*$^{RNAi}$ phenotypic scoring (Fig. 2e), serosal cuticle presence/absence was determined by piercing the fixed, dechorionated egg with a disposable needle (Braun Sterican 23 G, 0.60 × 25 mm): mechanically resistant eggs were scored for presence of the serosal cuticle while soft eggs that collapsed lacked serosal cuticle.

For *Tc-zen2*$^{RNAi}$ phenotypic scoring, larval cuticle preparations (Fig. 2c′, d′, f; Supplementary Fig. 3) were produced as previously described[26].

**Histology: cryosectioning, transcript and protein detection**. Whole mount in situ hybridization was performed as described[31], with probes synthesized from gene-specific primers (Supplementary Table 3) and colorimetric detection with NBT/BCIP. Embryos from all samples within the same experiment were treated in an identical fashion (e.g., duration of staining during NBT/BCIP precipitation). Specimens were imaged in Vectashield mountant with DAPI (Vector Laboratories) for nuclear counterstaining. Images were acquired on an Axio Plan 2 microscope (Zeiss). Image projections were generated with AxioVision (Zeiss) and Heli-conFocus 6.7.1 (Helicon Soft).

For cryosectioning, embryos were embedded in liquid sucrose–agarose embedding medium (15% sucrose, 2% agarose, [my-Budget Universal Agarose, Bio-Budget], PBS). Solid blocks of embedding medium containing embryos were stored overnight in 30% sucrose solution in PBS at 4 °C. The blocks were then embedded in Tissue Freezing Medium (Leica Biosystems) and flash-frozen in ice-cold isopentene (2-methylbutane). Samples were serially sectioned (20 μm, longitudinal; 30 μm, transverse) with a CM1850 cryostat (Leica Biosystems).

Protein was detected for both Tc-Zen1 and Tc-Zen2 with specific peptide antibodies (gift from the laboratory of Michael Schoppmeier)[79]. Immunohistochemistry on whole mounts and on sectioned material was performed by washing the samples six times for 10 min in blocking solution (2% BSA, 1% NGS, 0.1% Tween-20, PBS) followed by overnight incubation with the first antibody (rabbit anti-Tc-Zen1 and anti-Tc-Zen2, 1:1000) at 4 °C. Next, the samples were washed six times for 10 min in the blocking solution, followed by incubation with the secondary antibody (anti-rabbit Alexa Fluor 488 conjugate, 1:400, Invitrogen) for 3 h at room temperature (RT). Last, the samples were washed six times for 10 min in the blocking solution. Samples were then mounted in Vectashield mountant with DAPI. Low magnification images were acquired with an Axio Imager 2 equipped with an ApoTome 2 (Zeiss) structured illumination module, and maximum intensity projections were generated with ZEN blue software (Zeiss). High magnification images were acquired with an LSM 700 confocal microscope (Zeiss), and the projections were generated with ZEN 2 black software (Zeiss).

**Western blots**. For each 2-h developmental interval, protein lysate was obtained by lysing whole eggs in RIPA buffer (150 mM NaCl, 1.0% IGEPAL® CA-630, 0.5% sodium deoxycholate, 0.1% SDS, 50 mM Tris, pH 8.0; Sigma-Aldrich #R0278), followed by centrifugation. For each sample, 50 μg of protein extract was separated by SDS-PAGE. The PageRuler Prestained Protein Ladder (10–180 kDa; Thermo Fisher Scientific) was used as a size standard. Separated proteins were transferred onto nitrocellulose membrane (Thermo Fisher Scientific), which was blocked for 1 h in the blocking solution (100 mM Tris, 150 mM NaCl, pH 7.5, 0.1% Tween-20, 3% milk powder [Bebivita, Anfangsmilch]). Next, the membrane was incubated overnight at 4 °C with the first antibody (rabbit anti-Tc-Zen1 and anti-Tc-Zen2, 1:1000; mouse anti-Tubulin [Sigma-Aldrich #T7451: Monoclonal antiacetylated tubulin], 1:10,000). Afterwards, the membrane was washed three times for 10 min with the blocking solution at RT. The membrane was then incubated with the secondary antibodies (anti-rabbit and anti-mouse, HRP, 1:10,000, Novex) for 1 h at RT. Last, after the membrane was washed three times for 10 min with the blocking solution at RT, the membrane was incubated with ECL substrate according to the manufacturer's protocol (WesternSure ECL Substrate, LI-COR) and digital detection was performed on a western blot developing machine (C-DIGIT, LI-COR) with the high sensitivity settings. While the predicted molecular weights are 28.4 kDa for Tc-Zen1 and 33.7 kDa for Tc-Zen2 (ExPASy Mw Tool[80], last accessed 30 March 2020), in our hands we consistently observed migration of our samples

with actual sizes of ~39 kDa for Tc-Zen1 and ~42 kDa for Tc-Zen2, which may reflect posttranslational modifications.

**RNA-sequencing after RNAi**. RNAi for RNA-seq used the short, paralogue-specific dsRNAs (depicted schematically in Fig. 1c). For transcriptomic profiling by RNA-seq, three separate *Tc-zen1*<sup>RNAi</sup> experiments were conducted, each with a separate cohort of injected females. A total of seven *Tc-zen2*<sup>RNAi</sup> experiments were conducted: one for each biological replicate at each developmental stage. Samples chosen for sequencing were assessed by RT-qPCR for level of knockdown in RNAi samples, with *Tc-zen1* reduced to ~10% of wild-type levels and *Tc-zen2* to ~24% across biological replicates. For early development (6–14 hAEL), three biological replicates were sequenced for each experimental treatment, with 100-bp paired end reads on an Illumina HiSeq2000 machine. For late development (48–56 hAEL), four biological replicates were sequenced with 75-bp paired end reads on a HiSeq4000 machine. All sequencing was performed at the Cologne Center for Genomics, with six (HiSeq2000) or eight (HiSeq4000) multiplexed samples per lane yielding ≥6.6 Gbp per sample. The complete dataset of all 56 RNA-seq samples is deposited in GenBank (NCBI) under BioProject accession number PRJNA645519.

The quality of raw Illumina reads was examined with FastQC[81], and all RNA-seq samples were retained for analysis. The adapter sequences and low quality bases were removed with Trimmomatic v0.36[82]. Trimmomatic was also used to shorten 100-bp reads from the 3′ end to 75-bp reads to increase mapping efficiency (Supplementary Table 4)[83]. The overrepresented sequences of mitochondrial and ribosomal RNA were filtered out by mapping to a database of 1266 *T. castaneum* mitochondrial and ribosomal sequences extracted from the NCBI nucleotide database (accessed 21 October 2016, search query "tribolium [organism] AND (ribosomal OR mitochondrial OR mitochondrion) NOT (whole genome shotgun) NOT (Karroochloa purpurea)") with Bowtie2 v2.2.9[84]. Trimmed and filtered reads were mapped to the *T. castaneum* OGS3[67] (file name: Tcas5.2_GenBank.corrected_v5.renamed.mrna.fa) with RSEM[85]. The raw read count output from RSEM was compiled into count tables.

Both principal component and DE analyses were performed in R using the package DESeq2 v1.14.1[86] with default parameters. For PCA, raw (unfiltered) read counts were used. For DE analyses, to eliminate noise all genes with very low read counts were filtered out by sorting in Microsoft Excel, following recommendations[87]. Specifically, genes were excluded from DE analysis if read counts ≤10 in ≥1 biological replicates for both the knockdown and wild-type samples. For serosal immune genes, a given gene was considered to be expressed in the late serosa if we detected ≥100 RNA-seq reads in each of the four biological replicates for our wild-type samples. Throughout, our reporting of "DE genes" refers to analyses across all isoforms (18,536 isoform models) in the *T. castaneum* official gene set OGS3[67].

**Gene ontology (GO) analyses**. GO enrichment analysis was performed with Blast2GO[88] using two-tailed Fisher's exact test with a threshold false discovery rate of 0.05.

GO term analysis was performed by Blast2GO against the *Drosophila* database (accessed 9 June 2017). Only GO terms from the level 5 were considered. Next, GO terms were grouped into categories of interest based on similarity in function (Supplementary Table 1). Afterwards a unique count of *T. castaneum* gene sequences was calculated for each category of interest and the percentage was compared to the rest of the GO terms in the level 5 for each GO domain (Supplementary Fig. 8).

**Statistics and reproducibility**. All reported results were reproducible in our hands and consistent with published results with these methods and genes[12,13,26,35]. In addition to the use of robust biological replicates for any one technique, expression data were corroborated across RT-qPCR, RNA-seq, and in situ hybridization methods (mRNA transcript) or across western blot and immunohistochemistry (protein). RT-qPCR and RNA-seq analyses are based on 3–4 biological replicates, as indicated in the specific "Results" and "Methods" sections and associated figure legends (Figs. 1d, 2g, 3a and 6b; Supplementary Fig. 8B), and the source data values for these are available in Supplementary Data 5. Each biological replicate represents an independent sample and was derived from a different egg collection, with RNAi experiments conducted to obtain material from at least three distinct maternal cohorts. As noted above, RT-qPCR data were evaluated using LinRegPCR v12.16[75,76], and RNA-seq data were evaluated and processed with FastQC[81] and Trimmomatic v0.36[82]. To evaluate knockdown efficiency of long and short dsRNA fragments, beta regression analyses were performed with betareg v3.1-0[78], as noted above. DE statistical analyses were conducted with DESeq2 v1.14.1[86], and GO enrichment was determined with Blast2GO[88], as noted above.

**Reporting summary**. Further information on research design is available in the Nature Research Reporting Summary linked to this article.

## Data availability

All processed data and analyses generated during this study are included in this published article and its Supplementary Information files, including source data:

Supplementary Data 1–3: tables of differentially expressed genes from all comparisons; Supplementary Data 4: gene ontology (GO) terms for differentially expressed genes after *Tc-zen2* RNAi in late development; Supplementary Data 5: source values and dual plotting (means, individual values) for bar charts in figures (Figs. 1d, 2g, 3a and 6b; Supplementary Fig. 8B). The RNA-seq raw read data generated during the current study are available in GenBank (NCBI), under BioProject accession number PRJNA645519. The paralogue-specific peptide antibodies are available on request from the corresponding author or from the source laboratory[79].

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

## Acknowledgements

We thank Denise Mackrodt and Michael Schoppmeier for the kind gift of the Tc-Zen1 and Tc-Zen2 peptide antibodies, Viera Kovacova for bioinformatic program recommendations, Luigi Pontieri for assistance with statistical analyses, Thorsten Horn for sharing unpublished data on cuticle gene expression, Gustavo Lazzaro Rezende for discussions on cuticle regulation, and Hilary Ashe and Chris Rushlow for insights into *zen* regulation in *Drosophila*. RNA-seq analyses (trimming, filtering, OGS mapping) were performed on the Cologne High Efficient Operating Platform for Science (CHEOPS). We also thank Miltos Tsiantis and Siegfried Roth for helpful discussions and recommendations throughout the course of this research project. Siegfried Roth, Peter Heger, and Matthias Pechmann provided helpful feedback on the paper. This work was supported by funding from the German Research Foundation (Deutsche Forschungsgemeinschaft) through SFB 680 project A12 and Emmy Noether Program grant PA 2044/1-1 to K.A.P.

## Author contributions

D.G. designed experiments, collected and analyzed data, established the bioinformatic pipeline for the RNA-seq data, wrote the paper. I.M.V.J. analyzed data, established the bioinformatic pipeline for the RNA-seq data, edited the paper. K.A.P. conceived the project, designed experiments, analyzed data, established the bioinformatic pipeline for the RNA-seq data, wrote the paper.

## Funding

## Competing interests

The authors declare no competing interests.
