## [Peer Review File · Communications Biology]

Reviewers' comments:

Reviewer #2 (Remarks to the Author):

The study by Daniela et al. characterized the expression dynamics and transcriptional targets of the two Hox3/zen genes during embryogenesis in *Tribolium castaneum*. They identify a negative feedback loop between the two Tc-zen genes which promotes the developmental progression. They also found Tc-zen2 regulation was dynamic during the embryogenesis and showed the extraembryonic development is the major event within the transcriptional landscape of late embryogenesis. However, there are several critical technique flaws which could ruin the MS reliability, and those issues should be resolved before the consider acceptance of the manuscript for publication.

Major comments:

1. According to Fig1 D, although short fragment of zen1 dsRNA fragments did better work on avoiding cross-target effects to zen2, it still has a high efficiency on knocking down zen2 (relative expression level ~37.5% compared to control), which is already sufficient to affect the function of zen2. Likewise, data did not shown about the expression level about zen1 after zen2 RNAi. And a reasonable explanation is needed for the clear functional differentiation of the two genes under the presence of cross interference.
2. To elucidate the evolutionary history of zen functional changes, the author analysed the sequence conservation of the zen genes among the four *Tribolium* congenerics and found the tandem duplication of zen was conserved across the four related congenerics. To fully elucidate this problem, I would suggest the author present the phylogenetic tree of zen in more insect species and clarify when and why the tandem duplication occurred in *Tribolium*.
3. In the RNAi experiment, the dsRNA of GFP or other exogenous genes should be applied as control group. Wild type is not strong enough for gene function identification.
4. The methodology should be described more clearly. For the RNA-sequencing samples collection, please give the reason of using long dsRNAs for RNAi again. And which samples have been used finally for sequencing and data analysis should be pointed out. Also here the RNAi efficiency result should be provided containing the cross-target effect detection. And for early development (6-14 hAEL) and late development (48-56 hAEL), please explain why different biological replications have been carried out.

Minor comments:

1. Line 146 "As both paralogues are strongly expressed in early development (Fig. 2G)", I think this statement is not precisely, since it is obviously illustrated that during the decrease stage of zen1, the zen2 was gradually increased its expression at primitive pit stage (Fig. 2G) .
2. In the first paragraph, it mentioned "the Hox3 genes in winged insects, known as zen", please keep consistent to use "zen " or "Hox3" in the text. For example, Line 75, "insect Hox3 duplicates outside of *Drosophila*"
3. Line 33, Line 392, Line 437, Line 441, Line 444 and so on, "Hox" described as genes should be italic.

Reviewer #3 (Remarks to the Author):

This comprehensive analysis of Zen regulation and function reveals new regulatory interactions

contributing to extraembryonic membrane development.

The detailed analysis provides a deeper understanding of the specification vs morphogenesis of the serosa in a holometabolous insect. These findings are relevant to the study of extraembryonic membranes in other arthropods.

There are several issues that need addressing.

Figure 4 and Figure S3A and B what are the numbers along the left-hand side? If they are kDa, then they don't seem to match the legend or the expected sizes of Zen1 28.4 kDa and Zen2 33.7. What are the multiple bands in each western between 35- 40 kDa

For example. In Figure S3A, there is no band at 50 kDa, so what is the band at 40 kDa and where is zen1? Is the red arrow in Figure 3B pointing to a band at 40kDa? What is the band around 38 kDa?? Which protein ladder was used? Please give reference for protein extraction.

Is there a reference describing the specificity of the zen antibodies?

Line 112 it looks like zen2 expression is significantly knocked down by either the long or short zen1 dsRNA (figure 1d). How then does using the short zen1 make the effect gene specific and avoid off target effects on zen2 expression?

Line 234 wild type expression.... Do they mean expression of zen1 in wildtype?? There is no mutant zen in these experiments.

Line 221 do they mean possessive or plural of Cpr??

Line 353 does this require a data not shown indication?? And why isn't the data provided? If it is relevant enough to describe, it should be included.

Response to reviews on manuscript COMMSBIO-19-1293A

Overview:

We thank the editor and reviewers for the feedback and opportunity to revise the manuscript. Largely, there had been shared confusion by the reviewers in disentangling genetic regulation from dsRNA specificity. Briefly, Tc-Zen1 specifies the tissue domain for and activates *Tc-zen2*, such that loss of *Tc-zen1* necessarily reduces *Tc-zen2* expression. Also, we had not previously provided full documentation on antibody specificity. We apologize for this oversight on our part. We have amended the text for clarity on both issues and provided additional supplementary data (three figures). We also clarify confusion on our RNA-seq treatments and justify our experimental design, including controls and validation. Reviewer comments are reproduced in full (blue, italicized text) and interleaved with our responses and account of revisions (black text).

Reviewers' comments:

Reviewer #2 (Remarks to the Author):

The study by Daniela et al. characterized the expression dynamics and transcriptional targets of the two Hox3/zen genes during embryogenesis in Tribolium castaneum. They identify a negative feedback loop between the two Tc-zen genes which promotes the developmental progression. They also found Tc-zen2 regulation was dynamic during the embryogenesis and

showed the extraembryonic development is the major event within the transcriptional landscape of late embryogenesis. However, there are several critical technique flaws which could ruin the MS reliability, and those issues should be resolved before the consider acceptance of the manuscript for publication.

Major comments:

1. According to Fig1 D, although short fragment of zen1 dsRNA fragments did better work on avoiding cross-target effects to zen2, it still has a high efficiency on knocking down zen2 (relative expression level ~37.5% compared to control), which is already sufficient to affect the function of zen2.

We thank the reviewer for their scrutiny of our experimental design, as paralogue specificity is indeed critical. At the same time, the key finding of a negative feedback loop and the regulatory interactions between the paralogues underpins the complex genetic interactions that are the heart of this paper. As there was some confusion on this issue for both reviewers, we provide a detailed explanation here as well as in the indicated amendments to the manuscript.

Crucially, it was already known that *Tc-zen1* is essential to specify serosal tissue identity, and previous work also showed that *Tc-zen2* transcript is expressed in the serosa (van der Zee et al. 2005; manuscript reference 26). We make repeated mention that Tc-Zen1 specifies the serosa (unchanged: introduction lines 67-68, results lines 186-187 and 213-214; newly added: lines 131-132), and we show *Tc-zen2* expression strictly within the serosa (Figs. 3-4).

Thus, *Tc-zen2* expression will necessarily be reduced after *Tc-zen1* RNAi due to the loss of the tissue domain in which it should be expressed (as shown schematically in Fig. 2B). We go further in the new work to suggest that *Tc-zen2* is also a direct transcriptional target of Tc-Zen1 (the first, activating aspect of the regulatory feedback loop: first paragraph of the results section “The *Tc-zen* paralogues are mutual regulatory targets”). This strengthens the expectation of the necessary loss of *Tc-zen2* expression after *Tc-zen1* RNAi due to Tc-Zen1’s specification function and genetic regulation of *Tc-zen2*, not because of off-target RNAi knockdown. The original characterization of the paralogues states, “the early expression of *Tc-zen2* is dependent on *Tc-zen1* because no early *Tc-zen2* transcripts were found after *Tc-zen1* RNAi” (van der Zee et al. 2005). Here, we state (lines 221-223, unchanged): “Tc-Zen1 as a serosal specifier upregulates factors for definitive tissue differentiation, including *Tc-zen2* as a candidate (Fig. 6I).”

Our empirical results use higher sensitivity methods than the 2005 *in situ* assay, and curiously we actually detect more remaining *Tc-zen2* expression than would be expected *a priori* based on Tc-Zen1’s role. If Tc-Zen1 would be the sole factor for serosal identity and *Tc-zen2* expression, our high efficiency *Tc-zen1* RNAi (89-90% reduction in *Tc-zen1* expression [Fig. 1D] and 98.8% phenotypic penetrance [Fig. 2E]) should lead to the complete abolition of all detectable *Tc-zen2* expression. Instead, as noted by the reviewer, with the short, sequence-specific dsRNA fragment it is reduced but only to 35% of wild type levels,

not abolished (Fig. 1D). This in fact implies additional, Tc-Zen1-independent regulatory inputs for the initial activation of *Tc-zen2*.

We have amended the manuscript to clarify this issue in revised text (highlighted) in the initial results section (lines 109-116) and in the discussion (lines 366-370). We also do acknowledge the complexity of this regulatory situation later in the discussion (lines 396-398 and surrounding text, unchanged: “However, consistently lower knockdown efficiency for *Tc-zen2* than for *Tc-zen1* (Fig. 2, ²⁶) may reflect a dose-limiting lack of regulatory disentanglement.”).

[1b] Likewise, data did not shown about the expression level about zen1 after zen2 RNAi.

This must be an oversight. While only the aspect of homeodomain-spanning dsRNA is presented in Figure 1, to clarify our experimental design, the full assays (RT-qPCR, RNA-seq, and *in situ* hybridization) for *Tc-zen1* expression after *Tc-zen2* RNAi are shown in Figure 6. This is the core of our evidence arguing for the second aspect of the negative feedback loop: that Tc-Zen2 represses *Tc-zen1*. Thus, these results are built into our primary take-home message results and summary schematic figure.

[1c] And a reasonable explanation is needed for the clear functional differentiation of the two genes under the presence of cross interference.

In light of our comments above we would argue that there is not “cross interference” as the reviewer initially had supposed. The paralogues’ distinct functions have been demonstrated empirically in multiple publications from multiple laboratories on the RNAi phenotypes (Fig. 2A-D and the associated results text and references cited therein). We note that previous work obtained these results despite in some cases having used longer dsRNA fragments that had caused partial cross-knockdown. Indeed it is striking that these paralogues are so substantially diverged in function, an issue addressed in the first two paragraphs of the discussion.

2. To elucidate the evolutionary history of zen functional changes, the author analysed the sequence conservation of the zen genes among the four Tribolium congenerics and found the tandem duplication of zen was conserved across the four related congenerics. To fully elucidate this problem, I would suggest the author present the phylogenetic tree of zen in more insect species and clarify when and why the tandem duplication occurred in Tribolium.

We appreciate the reviewer’s interest in this aspect of paralogue evolutionary history. The phylogeny presented in Figure 1B is indeed a concise overview of relationships based on *zen* genes with clear extraembryonic functions. We have now expanded the results shown, with additional phylogenies in a new supplementary Figure S2 that includes expanded taxonomic

sampling and phylogenetic methods in support of the original conclusion. These trees represent the analysis approach already described in the manuscript's methods and slightly updated in revision (lines 509-516: "The same topology and comparable support values were also obtained with additional sequences and other methods (Fig. S2). This includes *Drosophila Zen* and/or *Z2* and/or *Drosophila* and *Megaselia Bicoid*, and/or insect *Zen* proteins with known expression but uncharacterized function^{36,59}, and bioinformatic predictions of *Zen* proteins in newly sequenced genomes. This also holds for trees generated with Bayesian methods from the same interface (MrBayes program v3.2.6, 1000 generations, 100 burn-in trees). That is, the *Tribolium Zen* proteins form a clear clade with various *Zen* outgroups, and then with the fly proteins as long branch outgroups."). Furthermore, the legend to the new supplementary Figure S2 notes, "In total, 38 trees with different parameters and taxa (30 proteins from 21 arthropod species) were analyzed, with consistent results."

For better resolution as to when the *Tribolium* duplication occurred, in Figure S2 we included *Zen* protein sequences from two distantly related beetles with recently sequenced genomes: the Colorado potato beetle (*Leptinotarsa decemlineata*) and the Asian longhorned beetle (*Anoplophora glabripennis*), where the latter does also – independently – have two copies of *zen*. However, fuller sampling such as this is mostly limited to bioinformatic predictions in *de novo* genomes. [REDACTION]. Even with additional sequenced genomes, the ultimate question the reviewer poses of why the duplication occurred remains an aspect for evolutionary speculation that cannot be addressed with such data.

3. In the RNAi experiment, the dsRNA of GFP or other exogenous genes should be applied as control group. Wild type is not strong enough for gene function identification.

We appreciate the importance of a valid negative control for RNAi experiments. However, both the *Tc-zen1* and *Tc-zen2* phenotypes are well established in the literature and explicitly confirmed here at the level of morphology and candidate gene expression analyses (Fig. 2). Please also note that the scale of injections required for these experiments, in order to obtain sufficient staged embryonic material for sequencing, makes the further injection of a negative control (instead of the uninjected wild type used here) untenable.

However, we are happy to reassure the reviewer that our experimental design reflects established, vetted approaches based specifically on previous work that also focused on *T. castaneum* pRNAi and RNA-seq analysis in the same beetle strain. In Stappert et al 2016 (Development 143: 2443-2454, doi:10.1242/dev.130641), both wild type and *dsRed* dsRNA-injected treatments were used as negative controls, and there was no difference between these samples at the level of global gene expression. In fact, the underlying Ph.D. thesis for that publication (p. 54 of Stappert, Dominik (2014). Two novel, complementary next generation sequencing approaches to reveal the dorso-ventral gene regulatory network of *Tribolium castaneum*. PhD Thesis, Universität zu Köln. <https://kups.ub.uni-koeln.de/5620/>) states:

“First, the transcriptomes of embryos derived from untreated wild type beetles were compared to transcriptomes of embryos derived from dsRed dsRNA injected mothers. No genes were found to be significantly differentially expressed. This suggest that the parental RNAi knockdown procedure does not cause expression changes (e.g. stress response) in the offspring of treated beetles, other than the ones specific to the gene knockdown. Hence, embryos from untreated wild type adults are sufficient as negative control and no mock injections have to be performed in future RNA-seq experiments.” This study’s focus on 7.5-11.5 hAEL falls within the same interval as our early RNA-seq experiment at 6-14 hAEL. We also note that an uninjected wild type control was used in the same *T. castaneum* strain in Oberhofer et al 2014 (Development 141: 1-11, doi:10.1242/dev.112797), where this latter study also was focusing on analysis of genes with previously documented RNAi phenotypes.

Furthermore, please note that all specific results reported here are validated or corroborated by multiple lines of evidence (e.g., RT-qPCR, *in situ* hybridization, and comparison to published GRN literature: as in Figs. 6, S6, and S8) that go beyond the initial differential expression calls from RNA-seq alone.

4. The methodology should be described more clearly. For the RNA-sequencing samples collection, please give the reason of using long dsRNAs for RNAi again.

We apologize for the ambiguous wording in the methods text, which reported the total of all RNAi experiments rather than those used specifically for RNA-seq. The statement in the results section is correct (lines 117-119, unchanged): “For all subsequent paralogue-specific functional testing, we thus designed our dsRNA fragments to exclude the homeobox and thereby avoid off-target effects (Fig. 1C: *Tc-zen1* short fragment: yellow; *Tc-zen2*: green).” The “long fragment” samples were used to assess the cross-target effect only (analysis presented in Fig. 1D), and since the figure legend explicitly states that this analysis involved three biological replicates, we have removed all mention of this preliminary analysis from the “RNA-sequencing after RNAi” methods section. The initial two sentences of that methods text have been revised accordingly (lines 604-606): “RNAi for RNA-seq used the short, paralogue-specific dsRNAs (depicted schematically in Fig. 1C). For transcriptomic profiling by RNA-seq, three separate *Tc-zen1*^{RNAi} experiments (each with a separate population of injected females) were conducted.”

[4b] And which samples have been used finally for sequencing and data analysis should be pointed out.

In the following lines of same methods section (lines 608-610, unchanged), we state: “Samples chosen for sequencing were assessed by RT-qPCR for level of knockdown in RNAi samples, with *Tc-zen1* reduced to ~10% of wild type levels and *Tc-zen2* to ~24% across biological replicates.” For clarity, we have now added the new comment “and all RNA-seq samples were retained for analysis” (lines 618-619).

[4c] Also here the RNAi efficiency result should be provided containing the cross-target effect detection.

These samples all used the short, paralogue-specific dsRNAs, as clarified in the amended methods text. The data on cross-paralogue effects are the central part of the analysis of the feedback loop, with Tc-Zen1 activation of *Tc-zen2*, followed by Tc-Zen2 repression of *Tc-zen1*. This is presented in Figure 6 (RT-qPCR and RNA-seq data from all samples included in the analyses) and the main text results in the final paragraph under the subheading “The *Tc-zen* paralogues are mutual regulatory targets”.

[4d] And for early development (6-14 hAEL) and late development (48-56 hAEL), please explain why different biological replications have been carried out.

The reviewer asks why three biological replicates were used for early development while four were used for late development. Briefly, three is of course the minimum sample size for statistical evaluation and is established practice in the field (e.g., Oberhofer et al 2014. Development 141: 1-11, doi:10.1242/dev.112797; Stappert et al 2016. Development 143: 2443-2454, doi:10.1242/dev.130641). Also, our experiment on early development was conducted first, when sequencing with a Hi-Seq 2000 machine was available. Reduced costs of using the newly available HiSeq 4000 allowed us to add an additional replicate when we subsequently performed the late development experiment. For late development, we also chose to add an additional replicate to slightly improve statistical sensitivity and robustness, as we were aware that the extraembryonic tissue offers a rare cell type within older whole-egg RNA-seq samples (mentioned in the results, lines 290-293, unchanged). While our bioinformatician colleagues have informed us that there is no upper limit on the number of biological replicates preferred for statistical power, for our experimental design and finely staged egg collections, four replicates was feasible in terms of generating sufficient yields of material for each replicate.

Minor comments:

1. Line 146 “As both paralogues are strongly expressed in early development (Fig. 2G)”, I think this statement is not precisely, since it is obviously illustrated that during the decrease stage of zen1, the zen2 was gradually increased its expression at primitive pit stage (Fig. 2G).

Our initial RT-qPCR survey, which is presented in Figure 2G, is a broadly staged overview across embryogenesis, and here early development encompasses the full range 6-14 hAEL, directly compared in this analysis to late development at 42-52 hAEL. For clarity, we have now explicitly added these age ranges to the figure legend text (line 718). This was deliberately organized to reflect the full breadth of “early” and “late” as it related to the phenotypically relevant stages presented in the preceding panels of that figure, and for which the qualitative developmental stages are indicated in panel 2G (e.g., “BF”, “DB”, etc.) and

defined in the figure legend. Indeed in the very next figure, we proceed to the fine-scale analysis with two-hour developmental resolution, in Figure 3 and the immediately following results section, where we do state (lines 152-153, unchanged): “As both paralogues are strongly expressed in early development (Fig. 2G), we first examined these stages in detail.”

2. In the first paragraph, it mentioned “the Hox3 genes in winged insects, known as zen”, please keep consistent to use “zen ” or “Hox3” in the text. For example, Line 75, “insect Hox3 duplicates outside of Drosophila”

We thank the reviewer for help with consistency and have amended this accordingly (now line 76).

3. Line 33, Line 392, Line 437, Line 441, Line 444 and so on, “Hox” described as genes should be italic.

We disagree. In these instances we are referring to the Hox genes as a class, not to specific genes, and therefore do not italicize these instances. For both of these minor comments, we systematically inspected all instances of “Hox” in the text to check for appropriate italicization.

Reviewer #3 (Remarks to the Author):

This comprehensive analysis of Zen regulation and function reveals new regulatory interactions contributing to extraembryonic membrane development.

The detailed analysis provides a deeper understanding of the specification vs morphogenesis of the serosa in a holometabolous insect. These finding are relevant to the study of extraembryonic membranes in other arthropods.

There are several issues that need addressed.

Figure 4 and Figure S3A and B what are the numbers along the lefthand side?

If they are kDa, then they don't seem to match the legend or the expected sizes of Zen1 28.4 kDa and Zen2 33.7. what are the multiple bands in each western between 35- 40 kDa

For example. In Figure S3A, there is no band at 50 kDa, so what is the band at 40 kDa and where is zen1? Is the red arrow in Figure 3B pointing to a band at 40kDa? What is the band around 38 kDa??

Which protein ladder was used? Please give reference for protein extraction.

Is there a reference describing the specificity of the zen antibodies?

We thank the reviewer for the careful scrutiny of our western blots. It may help if we respond to these questions in reverse order. Firstly, there is indeed a reference describing these peptide antibodies, in a Ph.D. thesis that is publicly available on-line and is cited in the

manuscript as reference 77 (in German: <https://opus4.kobv.de/opus4-fau/frontdoor/deliver/index/docId/6918/file/DeniseMackrodtDissertation.pdf>). As we also used the antibodies for immunohistochemistry, we mention this in the methods section on “histology”, immediately preceding the methods section on “western blots”, and this is now cross-referenced in the latter section (line 590). Specifically, we write (lines 565-566, unchanged): “Protein was detected for both Tc-Zen1 and Tc-Zen2 with specific peptide antibodies (gift from the laboratory of Michael Schoppmeier)⁷⁷.” Within that Ph.D. thesis (ref. 77), their figures 10 and 11 (PDF pages 49 and 51 as written on the pages; pages 63 and 65 of the digital document) show immunohistochemistry for Tc-Zen1 and Tc-Zen2, respectively, in wild type before going on to use these as markers in a variety of functional experiments that further corroborate their specificity. Our main text focus exclusively on Tc-Zen2 was remiss, and we have now added our own immunohistochemistry whole mount images at young stages for both Tc-Zen1 and Tc-Zen2 as a new supplementary Figure S5. The data in Figure S5 reproduce the results of that Ph.D. thesis and are also consistent with the spatiotemporal dynamics we report for the paralogue-specific transcript expression patterns (main text Figure 3).

Our study presents the first use of these antibodies for western blots. Firstly, yes our ladder sizes are in kDa, and this information has now been added in the main text methods (lines 584-586), Figure 4A legend, and Figure S4 (previously Figure S3) legend. We have also added a note on protein extraction (lines 581-583), which simply involves smashing eggs in RIPA buffer from Sigma (no formal citation).

Secondly, we agree with the reviewer that our actual band sizes do not migrate at the theoretically predicted sizes of these proteins, with both appearing larger than predicted. We have thus removed the predicted sizes from the figure legends, as this is unhelpful. Instead, within the “western blots” methods section we have added the explanatory comment (lines 598-601): “While the predicted molecular weights are 28.4 kDa for Tc-Zen1 and 33.7 kDa for Tc-Zen2 (ExpASY Mw Tool⁷⁸, last accessed 30 March 2020), in our hands we consistently observed migration of our samples with actual sizes of approximately 39 kDa for Tc-Zen1 and 42 kDa for Tc-Zen2, which may reflect post-translational modifications.”

Thirdly, we agree that the Zen-specific bands can appear as multi-band doublets, or with additional non-specific bands appearing in samples, especially with the co-detected anti-Tubulin control. This is not uncommon in our experience with western blot detection of multiple, diverse proteins, and we can assure the reviewer that the results we report here are highly robust and reflect numerous western blot assays of the Tc-Zen paralogues. To further document this, we have now expanded Figure S4 to include additional panels of additional blots, taking care to reproduce the uncropped blots, warts and all and yet in fact very clean for westerns. The original panels of this figure (former Figure S3) are retained: original panel S3A remains as panel S4A and is now shown uncropped and with supporting Ponceau S staining, while original panel S3B is now panel S4E and supported by an enlarged inset image in S4E'. These results demonstrate that when detected singly both paralogues are detected as single bands. Furthermore, for additional corroboration of specificity, in this figure we have now added blots demonstrating that the Tc-Zen2 band is specifically reduced in *Tc-zen2* RNAi samples compared to wild type samples. (This analysis was performed for

Tc-Zen2 only, as it was necessary to confirm that the knockdown effect persists until the late EE morphogenesis stages, when *Tc-zen2* transcript is not detected. As *Tc-zen1* transcript is expressed at the time of its known function and of our RNA-seq samples, in that case it was sufficient to confirm RNAi knockdown by RT-qPCR.) However, we are confused as to why in Figure S4A the reviewer would expect a band at 50 kDa and not recognize that the prominent bands at ~39 kDa are indeed for Tc-Zen1. We have explicitly stated in the revised figure legend that this blot and others, as indicated, did not include detection of anti-Tubulin at 59 kDa in the same lane. We can also confirm that indeed the red arrow in Figure S4E (previously S3B) does point to a band of >40 kDa for Tc-Zen2, as indicated in the figure legend.

Line 112 it looks like zen2 expression is significantly knocked down by either the long or short zen1 dsRNA (figure 1d). How then does using the short zen1 make the effect gene specific and avoid off target effects on zen2 expression?

Please see our first response to the other reviewer on this issue. As this reviewer recognizes, our study is on the complex regulatory interactions that govern extraembryonic membrane development, including the primary role of Tc-Zen1 in specifying the serosal tissue domain and activating *Tc-zen2*, such that these results reflect the genetic regulatory situation (short fragment) in addition to the potential for cross-paralogue knockdown (long fragment).

Line 234 wild type expression.... Do they mean expression of zen1 in wildtype?? There is no mutant zen in these experiments.

That is correct. We have amended the text to read “its wild type expression”, referring back to *Tc-zen1* as the subject of this sentence (now line 243).

Line 221 do they mean possessive or plural of Cpr??

We refer to the plural, but as “Cpr” is an established abbreviation in its own right, we did not wish to simply append an “s” without punctuation. For clarity, this is now amended to “Cpr proteins” (now line 230).

Line 353 does this require a data not shown indication?? And why isn't the data provided? If it is relevant enough to describe, it should be included.

This must be an oversight (coupled with potentially ambiguous punctuation of the referencing), as this refers to work by others in cited reference 39. We have amended these sentences to explicitly state “in a recent study³⁹” and to cite this reference twice (originally once) in this discussion paragraph (now lines 362 and 366) in addition to also citing it in the legend for Figure 1A (no change).

REVIEWERS' COMMENTS:

Reviewer #3 (Remarks to the Author):

The authors have provided a thorough response to my questions. I am satisfied with the rigor of the research and the extensive discussion of this novel negative regulation loop.

Reviewer #4 (Remarks to the Author):

I find the response of the authors to the criticism of reviewer 2 satisfactory and comprehensive. I suggest that the authors make the point even more clear by adding one sentence (see below)

In detail:

Reviewer 2 major comment 1a:

The reviewer refers to potential off-target effects between the knock-down of zen1 versus zen2. The authors make explicit note of one off-target stretch in the homeodomain and subsequently use the small fragments for their analyses. And they state explicitly:

"For all subsequent paralogue-specific functional testing, we thus designed our dsRNA fragments to exclude the homeobox and thereby avoid off-target effects..."

This approach avoids the OTE effects that the reviewer 2 is referring to. Hence, the criticism is appropriately answered.

Nevertheless, I suggest that the authors add a respective explicit statement "The alignment revealed that there was no ≥ 20 mer apart from the one found in the homeodomain. Hence, RNAi with the short fragments is paralogue specific while the long zen1 fragment has an off-target effect on zen2. For all subsequent..."

1b

Satisfactorily answered.

1c

Would be answered with the additional sentence referred to in 1a

2

Satisfactorily answered by adding additional analyses.

3

In the *Tribolium* field, these controls are not expected any more because with respect to embryogenesis we have many carefully analyzed examples where dsRNA treatment of other sequences did not lead to the phenotypes observed in zen1/zen1. Therefore, I agree with the authors that this experiment is not necessary.

4

Satisfactorily answered by changing the wording of the methods section.

4b

See 1.

4d

Satisfactorily answered.

Response to reviews on manuscript COMMSBIO-19-1293B

We thank the editor and reviewers for the feedback and opportunity to finalize revisions of our manuscript. There was one minor textual change suggested by one reviewer.

REVIEWERS' COMMENTS:

Reviewer #3 (Remarks to the Author):

The authors have provided a thorough response to my questions. I am satisfied with the rigor of the research and the extensive discussion of this novel negative regulation loop.

Reviewer #4 (Remarks to the Author):

I find the response of the authors to the criticism of reviewer 2 satisfactory and comprehensive. I suggest that the authors make the point even more clear by adding one sentence (see below)

In detail:

Reviewer 2 major comment 1a:

The reviewer refers to potential off-target effects between the knock-down of zen1 versus zen2. The authors make explicit note of one off-target stretch in the homeodomain and subsequently use the small fragments for their analyses. And they state explicitly:

“For all subsequent paralogue-specific functional testing, we thus designed our dsRNA fragments to exclude the homeobox and thereby avoid off-target effects...”

This approach avoids the OTE effects that the reviewer 2 is referring to. Hence, the criticism is appropriately answered.

Nevertheless, I suggest that the authors add a respective explicit statement “The alignment revealed that there was no ≥ 20 mer apart from the one found in the homeodomain. Hence, RNAi with the short fragments is paralogue specific while the long zen1 fragment has an off-target effect on zen2. For all subsequent...”

Authors' response:

We appreciate Reviewer 4's suggestions for making the issue of dsRNA design and paralogue specificity explicit. Accordingly, we have adjusted the sentence and added the highlighted and underlined text, which are directly supported by Figure 1C:

“For all subsequent functional testing we thus designed our dsRNA fragments to exclude the homeobox, thereby avoiding off-target effects **and ensuring paralogue-specific knockdown** (Fig. 1C: *Tc-zen1* short fragment: yellow; *Tc-zen2*: green; **no ≥ 20 -bp stretches of nucleotide identity in these regions**).”